# Beyond Higher Rank: Token-wise Input-Output Projections for Efficient Low-Rank Adaptation

**Shiwei Li[1 2] Xiandi Luo[1] Haozhao Wang[1*]Xing Tang[2]**
**Ziqiang Cui[3] Dugang Liu[4] Yuhua Li[1] Xiuqiang He[2*]Ruixuan Li[1*]**
[1]Huazhong University of Science and Technology [2]Shenzhen Technology University
[3]City University of Hong Kong [4]Shenzhen University
{lishiwei, togawa_sakiko, hz_wang, idcliyuhua, rxli}@hust.edu.cn
xing.tang@hotmail.com, ziqiang.cui@my.cityu.edu.hk
dugang.ldg@gmail.com, hexiuqiang@sztu.edu.cn

## Abstract

Low-rank adaptation (LoRA) is a parameter-efficient fine-tuning (PEFT) method widely used in large language models (LLMs). LoRA essentially describes the projection of an input space into a low-dimensional output space, with the dimensionality determined by the LoRA rank. In standard LoRA, all input tokens share the same weights and undergo an identical input-output projection. This limits LoRA's ability to capture token-specific information due to the inherent semantic differences among tokens. To address this limitation, we propose **Token-wise Projected Low-Rank Adaptation (TopLoRA)**, which dynamically adjusts LoRA weights according to the input token, thereby learning token-wise input-output projections in an end-to-end manner. Formally, the weights of TopLoRA can be expressed as $B\Sigma_X A$, where $A$ and $B$ are low-rank matrices (as in standard LoRA), and $\Sigma_X$ is a diagonal matrix generated from each input token $X$. Notably, TopLoRA does not increase the rank of LoRA weights but achieves more granular adaptation by learning token-wise LoRA weights (i.e., token-wise input-output projections). Extensive experiments across multiple models and datasets demonstrate that TopLoRA consistently outperforms LoRA and its variants. The code is available at `https://github.com/Leopold1423/toplora-neurips25`.

## 1 Introduction

Recent advancements in pretrained large language models (LLMs) have led to significant improvements in a variety of natural language processing and vision tasks [3, 47, 40, 2]. Traditionally, these models require full fine-tuning (FFT) to update all their parameters for specific downstream tasks. However, due to the large size of pretrained models, FFT can be computationally expensive, particularly in resource-constrained environments. To address this challenge, parameter-efficient fine-tuning (PEFT) methods have been introduced to reduce the number of trainable parameters and decrease fine-tuning costs [14, 7, 38, 41]. Among these methods, low-rank adaptation (LoRA) [10] has emerged as one of the most widely used techniques. As shown in Figure 1(a), LoRA freezes the pretrained weight matrix $W \in \mathbb{R}^{m \times n}$ and learns two smaller low-rank matrices to approximate the weight update as $\Delta W = BA$, where $A \in \mathbb{R}^{r \times n}$, $B \in \mathbb{R}^{m \times r}$, and the LoRA rank $r \ll \min\{m, n\}$.

Despite its effectiveness, LoRA typically exhibits a performance gap when compared to FFT, often attributed to the limited number of trainable parameters [10, 27]. Previous studies have also shown that increasing the LoRA rank generally improves fine-tuning performance [35, 26]. Recently,

---

*Corresponding authors.

39th Conference on Neural Information Processing Systems (NeurIPS 2025).

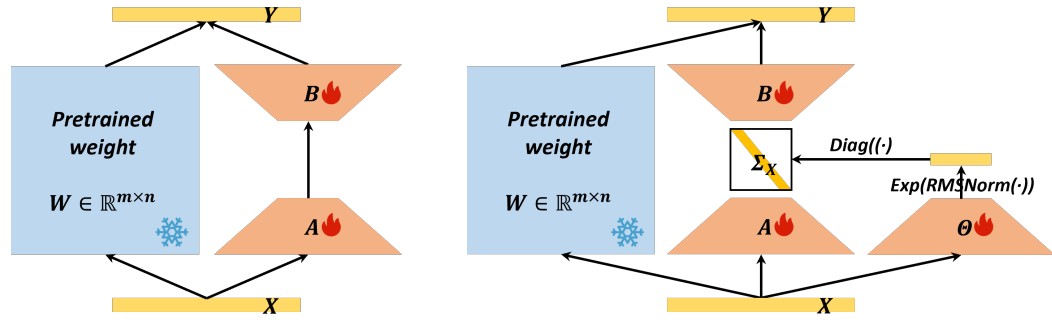

|  (a) Low-Rank Adaptation (LoRA)  |  (b) Token-wise Projected Low-Rank Adaptation (TopLoRA)  |

Figure 1: An illustration of TopLoRA in comparison to LoRA [10]. TopLoRA additionally learns a projector $\Theta$ to generate a diagonal matrix $\Sigma_X$ based on the input token $X$, which is then used to adjust the LoRA weights (i.e., the input-output projections) for each token. The operators $\text{Exp}(\cdot)$ and $\text{RMSNorm}(\cdot)$ refer to the exponential function and root mean square normalization [49], respectively.

Zeng and Lee [48] explored the expressive power of LoRA, using the LoRA rank to quantify the approximation errors between the LoRA weights ($\Delta W = BA$) and the assumed optimal weight update. Their results suggest that smaller LoRA ranks lead to larger approximation errors. However, a more intuitive explanation of how the LoRA rank affects fine-tuning performance remains lacking. This raises the question of *whether there is a more intuitive way to understand the role of LoRA rank and whether other factors, beyond LoRA rank, also influence fine-tuning performance*.

To answer this question, we explore the impact of LoRA rank from the perspective of input-output projections. Specifically, the matrix $B$ can be expressed using QR decomposition as $B = Q_B R_B \in \mathbb{R}^{m \times r}$, where $Q_B = [Q_B^1, \ldots, Q_B^r] \in \mathbb{R}^{m \times r}$ is a semi-unitary matrix (i.e., $Q_B^\top Q_B = I$) and $R_B \in \mathbb{R}^{r \times r}$ is a right triangular matrix. Similarly, the matrix $A$ can be expressed using LQ decomposition as $A = L_A Q_A \in \mathbb{R}^{r \times n}$, where $Q_A = [Q_A^1, \ldots, Q_A^r]^\top \in \mathbb{R}^{r \times n}$ is also a unitary matrix (i.e., $Q_A Q_A^\top = I$) and $L_A \in \mathbb{R}^{r \times r}$ is a left triangular matrix. At this point, the LoRA weights can be represented as follows:

$$\Delta W = BA = Q_B(R_B L_A)Q_A = Q_B P Q_A = [Q_B^1, \ldots, Q_B^r][P_1, \ldots, P_r][Q_A^1, \ldots, Q_A^r]^\top. \quad (1)$$

As shown, the LoRA weights consist of three components: the input space $Q_A = [Q_A^1, \ldots, Q_A^r]^\top$, the output space $Q_B = [Q_B^1, \ldots, Q_B^r]$, and the input-output projection $P = R_B L_A = [P_1, \ldots, P_r] \in \mathbb{R}^{r \times r}$. Notably, LoRA captures only the projection of the input token onto the column vectors of $Q_A$, with all other projections transformed to zero. Additionally, the LoRA output is constrained to a linear combination of the row vectors of $Q_B$. Thus, increasing the LoRA rank $r$ raises the dimensionality of both the input and output spaces, thereby improving LoRA's expressive power.

Based on the above analysis, we reveal another factor, beyond increasing the LoRA rank $r$, that can enhance its expressiveness. Specifically, different tokens exhibit distinct semantics and distributions, and the shared input-output projection ($P = R_B L_A$) is insufficient for capturing token-specific information. To address this limitation, we introduce **Token-wise Projected Low-Rank Adaptation (TopLoRA)**, which dynamically adjusts LoRA weights for each token, as shown in Figure 1(b). TopLoRA utilizes a projector, $\Theta$, to generate a diagonal matrix, $\Sigma_X$, based on the input token $X$. This matrix, in turn, adjusts the value of $P = R_B L_A$ to $P_X = R_B \Sigma_X L_A$. Importantly, the goal of TopLoRA is not to increase the rank of LoRA weights (i.e., the dimensionality of the input and output spaces). Instead, it focuses on capturing distinct key information from different tokens through token-wise input-output projections, enabling finer-grained adaptation even with a limited rank.

The main contributions can be summarized as follows:

- We analyze LoRA from the perspective of input and output projection and identify another factor, beyond the LoRA rank, that limits the expressiveness of LoRA: the shared input-output projection of LoRA is insufficient for capturing token-specific information.

- We propose TopLoRA, which enables LoRA to apply distinct input-output projections for different input tokens. By adjusting the LoRA weights with token-wise diagonal matrices, TopLoRA achieves finer-grained adaptation even with a limited rank.

- We conduct extensive experiments on various models and datasets, demonstrating that TopLoRA consistently outperforms LoRA and its variants. At the same rank, TopLoRA achieves a 2-3% accuracy improvement over LoRA.

## 2 Motivations

### 2.1 Low-Rank Adaptation

Building on the hypothesis that weight updates during fine-tuning exhibit low intrinsic dimensionality, Hu et al. [10] proposed Low-Rank Adaptation (LoRA), which updates pretrained weights using the product of two low-rank matrices. Given a pretrained weight matrix $W \in \mathbb{R}^{m \times n}$, the forward propagation of LoRA can be formulated as:

$$Y = (W + \Delta W)X = (W + \alpha/r BA)X, \tag{2}$$

where $B \in \mathbb{R}^{m \times r}$, $A \in \mathbb{R}^{r \times n}$, the LoRA rank $r \ll \min(m, n)$, and $\alpha$ is an adjustable scaling factor. For simplicity, we omit the coefficient $\alpha/r$ in the following discussion. During fine-tuning, the pretrained weights remain frozen, and only the low-rank matrices $A$ and $B$ are updated. Note that the LoRA rank $r$ determines the rank of the weight update matrix as $\text{rank}(\Delta W) = \text{rank}(BA) \leq \min\{\text{rank}(A), \text{rank}(B)\} \leq r$. A higher LoRA rank $r$ generally leads to better fine-tuning performance, but also increases the number of trainable parameters.

### 2.2 Input-Output Projections of Low-Rank Adaptation

Most existing studies acknowledge that the LoRA rank significantly affects fine-tuning performance but lack an intuitive explanation [35, 26, 12, 4]. In this paper, we provide a clearer interpretation of the role of LoRA rank from the perspective of input-output projections and explore additional factors that may influence LoRA's fine-tuning performance. As discussed in the previous section, applying LQ and QR decompositions to the matrices $A \in \mathbb{R}^{r \times n}$ and $B \in \mathbb{R}^{m \times r}$ yields:

$$A = L_A Q_A, \quad B = R_B Q_B, \tag{3}$$

where $Q_A = [Q_A^1, \ldots, Q_A^r]^\top \in \mathbb{R}^{r \times n}$ and $Q_B = [Q_B^1, \ldots, Q_B^r] \in \mathbb{R}^{m \times r}$ are unitary matrices (i.e. $Q_A^\top = Q_A^{-1}$ and $Q_B^\top = Q_B^{-1}$), $L_A \in \mathbb{R}^{r \times r}$ and $R_B \in \mathbb{R}^{r \times r}$ are left- and right-triangular matrices, respectively. Therefore, the LoRA weights can then be expressed as:

$$\Delta W = BA = Q_B(R_B L_A)Q_A = Q_B P Q_A = [Q_B^1, \ldots, Q_B^r][P_1, \ldots, P_r][Q_A^1, \ldots, Q_A^r]^\top, \tag{4}$$

where $P = R_B L_A = [P_1, \ldots, P_r] \in \mathbb{R}^{r \times r}$ represents the interaction between the matrices $Q_A$ and $Q_B$. Any input token $X \in \mathbb{R}^n$ can be decomposed into $X = \hat{X} + \sum_{i=1}^r \alpha_i Q_A^i$, where $\hat{X}$ denotes the component orthogonal to all $Q_A^i$, satisfying $Q_A \hat{X} = 0$. Thus, the LoRA output can be expressed as:

$$\Delta W X = Q_B P Q_A X = \sum_{i=1}^r \alpha_i Q_B P Q_A Q_A^i = \sum_{i=1}^r \alpha_i \sum_{j=1}^r P_{i,j} Q_B^j, \tag{5}$$

which indicates that LoRA captures only the component of the input token along $r$ directions $\{Q_A^1, \ldots, Q_A^r\}$ and maps them to $r$ corresponding output directions $\{Q_B^1, \ldots, Q_B^r\}$, where $P_{i,j}$ denotes the scaling factor from $Q_A^i$ to $Q_B^j$. Accordingly, the LoRA weights, as shown in Eq. (4), can be divided into three components: the input space $Q_A = [Q_A^1, \ldots, Q_A^r]$, which represents the input range that LoRA can effectively capture; the output space $Q_B = [Q_B^1, \ldots, Q_B^r]$, which represents the output range that LoRA can cover; and the input-output projection $P = R_B L_A = [P_1, \ldots, P_r]$, which models the scaling of each $Q_A^j$ as it transforms into each $Q_B^i$.

From this perspective, the LoRA rank determines the dimensionality of the input and output spaces. A lower rank restricts the model's ability to extract and process information from the input tokens, which intuitively explains why LoRA often performs worse than FFT and why increasing the rank typically enhances performance. Previous studies have mainly focused on increasing the rank of

LoRA weights to expand the dimensionality of both the input space ($Q_A$) and the output space ($Q_B$). For example, HiRA [12] and KronA [4] increase the rank of LoRA weights using the Hadamard and Kronecker products, respectively. MELoRA [35], on the other hand, achieves a higher rank by stacking low-rank matrices along the diagonal.

In contrast, we identify another important factor that affects the expressiveness of LoRA: the input-output projection (i.e., $P$). In standard LoRA, all tokens share the same input-output projection. However, since the semantics of different tokens vary significantly, even the same projection (e.g., $Q_A^i$) may represent different information for different tokens, requiring distinct processing. Existing methods that apply a uniform input-output projection across all tokens fail to capture token-specific information. This motivates us to explore a method that provides adaptive input-output projections tailored to individual tokens. In this way, we can increase the diversity of input-output projections, thereby enhancing the expressiveness of LoRA, even with smaller LoRA ranks.

## 3 Methodology

In this section, we introduce the **Token-wise Projected Low-Rank Adaptation (TopLoRA)**, a LoRA architecture designed to optimize the input-output projections at the token level.

### 3.1 Token-wise Projected Low-Rank Adaptation

Based on the above analysis, the optimization of LoRA seeks to identify a pair of input and output spaces along with the corresponding input-output projections. However, different tokens share the same LoRA weights and thus undergo identical input-output projections, which limits the ability to capture token-specific information. To overcome this limitation, we propose Token-wise Projected Low-Rank Adaptation (TopLoRA), which dynamically adjusts the LoRA weights for each token. To achieve this in a parameter-efficient manner, TopLoRA maintains an additional network, $\Gamma$, with parameters $\Theta$, which generates a diagonal matrix $\Sigma_X$ based on the input token $X$. This matrix is then used to modify the LoRA weight (i.e., the input-output projection) as follows:

$$\Delta W_X = B\Gamma_\Theta(X)A = B\Sigma_X A. \tag{6}$$

It is important to note that the goal of TopLoRA is not to increase the rank of the LoRA weights, but rather to find the optimal input-output projection for each token, even within a limited rank.

### 3.2 Optimization of Token-wise Diagonal Matrix

The effectiveness of TopLoRA depends on the careful design of its token-wise diagonal matrix, $\Sigma_X$, which is generated by a specialized network, $\Gamma_\Theta$. Notably, the parameters $\Theta$ employ Kaiming initialization [8], consistent with the initialization of matrix $A$, ensuring stable training when using a uniform learning rate for all parameters. Further, to facilitate effective learning of $\Sigma_X$, we apply root mean square normalization (RMSNorm) [49] followed by an exponential transformation on the network output, $\Theta X$, as shown below:

$$\Sigma_X = \text{Diag}(\text{Exp}(\text{RMSNorm}(\Theta X))). \tag{7}$$

RMSNorm ensures that the value of $\Gamma_X$ is not influenced by the magnitude of token $X$ or the projection parameter $\Theta$. It essentially broadens the distribution range of $\Sigma_X$, thus increasing the differences between the diagonal matrices of different tokens. The exponential transformation converts the zero-centered normalized values into strictly positive scaling factors. It prevents information loss from near-zero values of $\Theta X$ and ensures that even subtle but important variations in the normalized projections can be effectively captured.

### 3.3 Comparison with Low-Rank Adaptation

TopLoRA introduces a dynamic term, $\Sigma_X$, which enhances standard LoRA by enabling the explicit modeling of token-specific information. Specifically, the output of TopLoRA can be decomposed into two components as follows:

$$\Delta W_X X = B\Sigma_X AX = BAX + B(\Sigma_X - I)AX, \tag{8}$$

where $I$ denotes the identity matrix. The first term, $BAX$, represents the standard LoRA output, capturing global patterns across all input tokens. The second term, $B(\Sigma_X - I)AX$, introduces a dynamic, input-dependent adaptation mechanism that models fine-grained variations unique to each token. Here, $\Sigma_X$ serves as a learned gating mechanism that adjusts the importance of different feature dimensions for each token. Note that when $\Sigma_X = I$, TopLoRA reverts to standard LoRA. This architecture retains the advantages of low-rank parameterization while significantly enhancing LoRA's expressive power, particularly for tasks where tokens require distinct projections.

It is worth noting that MoELoRA [30] adopts the mixture-of-experts mechanism and achieves similar token adaptive weights. Specifically, MoELoRA trains multiple LoRA adapters $\{(A_i, B_i) \mid i > 1\}$ as experts, each specializing in distinct knowledge. It also maintains a routing network that dynamically combines the values of these adapters to generate adaptive weights. However, the motivation of this paper differs from that of MoELoRA. MoELoRA utilizes the mixture-of-experts architecture to cope with the knowledge diversity in training data and downstream tasks. Unlike MoELoRA, our analysis reveals that LoRA's expressiveness may be constrained by the shared input-output projection. TopLoRA overcomes this limitation more efficiently and directly. Moreover, the TopLoRA structure can be incorporated into each expert in MoELoRA to further enhance its expressiveness. In the following section, we will experimentally demonstrate the superior performance of TopLoRA compared to an improved MoELoRA method.

## 4    Experiments

### 4.1    General Settings

**Datasets and Models.**    In this section, we evaluate TopLoRA on three benchmarks using different model architectures. First, we assess TopLoRA's natural language understanding (NLU) capabilities on the GLUE benchmark [42], which includes eight sub-tasks. The models used are RoBERTa-Base and RoBERTa-Large [28]. Next, we examine TopLoRA's natural language generation (NLG) capabilities on two reasoning benchmarks compiled by Hu et al. [11]: mathematical reasoning and commonsense reasoning, which consist of 10k and 170k training samples, respectively. The mathematical reasoning benchmark comprises six sub-tasks, whereas the commonsense reasoning benchmark includes eight. To demonstrate TopLoRA's versatility, we evaluate it across multiple architectures and scales, including Gemma-7B [36], LLaMA-3-8B [5], and Qwen2.5-14B [47].

**Baseline Methods.**    The effectiveness of TopLoRA is demonstrated by comparison with several baseline methods, including LoRA [10] and its variants: DoRA [27], MELoRA [35], and HydraLoRA [37]. Each of these methods introduces distinct modifications to the original LoRA framework. For example, DoRA improves LoRA by decomposing pretrained weights into magnitude and direction components. The magnitude is learned through a trainable vector, while the direction is updated using LoRA. MELoRA enhances LoRA by incorporating mini-ensemble low-rank adapters, achieving a higher rank at a reduced parameter cost compared to the standard LoRA approach. HydraLoRA refines LoRA further with an asymmetric architecture that increases parameter efficiency. It employs a shared $A$ matrix and multiple $B$ matrices, which are dynamically routed to optimize performance. Essentially, HydraLoRA improves upon MoELoRA by facilitating better knowledge sharing and differentiation among the experts. These methods represent different strategies for improving LoRA, including better optimization, higher rank, and the mixture-of-experts architecture. By comparing with these techniques, we can clearly demonstrate the superiority of TopLoRA.

**General Hyperparammeters.**    In the main experiments, we evaluate the accuracy of LoRA across ranks 8, 16, and 32 to analyze the trade-off between the number of parameters and model performance. For TopLoRA, we test ranks 8 and 16. For the three LoRA variants, we adjust the number of trainable parameters to match those of LoRA with a rank of 16. Specifically, we set the rank of DoRA to 16, while HydraLoRA uses a rank of 8 with three $B$ matrices. For MELoRA, the Mini-LoRA rank is set to 16, with four Mini-LoRA groups. The general settings include the AdamW optimizer [29], a LoRA dropout rate of 0.05, and no weight decay. Each experiment is repeated three times, and the average results are reported. Further details can be found in Appendix A and B.

Table 1: The accuracy of different methods on **General Language Understanding** tasks with various pretrained models. The highest average precision is bolded, and the second-highest is underlined.

| | | #Params | RTE | MRPC | STS-B | CoLA | SST-2 | QNLI | MNLI | QQP | Avg |
|---|---|---|---|---|---|---|---|---|---|---|---|
| RoBERTa-Base | LoRA($r = 8$) | 0.29M | 72.56 | 87.25 | 87.12 | 56.10 | 93.46 | 91.58 | 84.89 | 87.46 | 82.55 |
| | LoRA($r = 16$) | 0.59M | 72.92 | 87.99 | 87.46 | 55.21 | 93.81 | 91.89 | 85.52 | 87.79 | 82.82 |
| | LoRA($r = 32$) | 1.18M | 75.09 | 89.22 | 88.01 | 58.58 | 93.58 | 90.12 | 85.84 | 88.37 | 83.60 |
| | DoRA($r = 16$) | 0.61M | 74.85 | 87.83 | 88.48 | 56.46 | 93.39 | 91.86 | 85.25 | 87.89 | 83.25 |
| | MELoRA($r = 16$) | 0.59M | 75.45 | 88.73 | 87.27 | 54.43 | 93.00 | 91.51 | 84.93 | 87.53 | 82.86 |
| | HydraLoRA($r = 8$) | 0.65M | 73.65 | 89.46 | 88.53 | 57.03 | 93.23 | 91.89 | 85.52 | 87.57 | 83.36 |
| | **TopLoRA**($r = 8$) | 0.44M | 78.34 | 87.99 | 90.05 | 58.55 | 92.43 | 92.02 | 85.60 | 88.12 | 84.14 |
| | **TopLoRA**($r = 16$) | 0.89M | 78.34 | 88.73 | 88.90 | 60.34 | 93.35 | 92.11 | 85.65 | 88.43 | **84.48** |
| RoBERTa-Large | LoRA($r = 8$) | 0.79M | 71.96 | 88.40 | 89.88 | 59.76 | 95.41 | 93.07 | 88.67 | 87.89 | 84.38 |
| | LoRA($r = 16$) | 1.57M | 77.74 | 88.48 | 90.60 | 61.23 | 95.57 | 93.72 | 89.29 | 88.25 | 85.61 |
| | LoRA($r = 32$) | 3.15M | 81.35 | 89.64 | 91.45 | 60.90 | 95.60 | 93.76 | 89.52 | 88.61 | 86.35 |
| | DoRA($r = 16$) | 1.62M | 80.14 | 88.73 | 90.94 | 61.73 | 95.68 | 93.45 | 89.30 | 88.29 | 86.03 |
| | MELoRA($r = 16$) | 1.57M | 79.48 | 87.75 | 90.17 | 60.59 | 95.87 | 93.21 | 88.86 | 87.89 | 85.48 |
| | HydraLoRA($r = 8$) | 1.72M | 79.42 | 89.46 | 90.63 | 61.07 | 95.76 | 93.39 | 89.25 | 88.26 | 85.91 |
| | **TopLoRA**($r = 8$) | 1.18M | 80.51 | 89.30 | 91.54 | 61.75 | 95.64 | 93.89 | 89.66 | 88.83 | 86.39 |
| | **TopLoRA**($r = 16$) | 2.36M | 85.20 | 90.44 | 91.50 | 64.56 | 95.64 | 94.20 | 89.94 | 88.94 | **87.55** |

## 4.2   Natural Language Understanding Tasks

**Implementation Details.**   The GLUE benchmark comprises eight sub-tasks, each with distinct training and test sets. For each sub-task, we fine-tuned the model on the training set and assessed its accuracy on the corresponding test set. The learning rates for RoBERTa-Base and RoBERTa-Large were set to 3e-4 and 1e-4, respectively. A warm-up ratio of 0.03 and linear learning rate decay were used. The number of training epochs varied across sub-tasks; further details are provided in Appendix A. TopLoRA and the baseline methods were only applied to the query and value weights.

**Results on the GLUE Tasks.**   As shown in Table 1, increasing the LoRA rank improves fine-tuning performance. Specifically, raising the rank from 8 to 32 increases the average accuracy of LoRA by 1.05% and 1.97% on RoBERTa-Base and RoBERTa-Large, respectively. Compared to LoRA ($r = 8$), TopLoRA ($r = 8$) achieves an accuracy improvement of 1.5% and 2.01%, surpassing LoRA ($r = 32$). Notably, TopLoRA's parameter count is only about 1.5 times that of LoRA at the same rank. The improvements of other baselines are relatively modest. DoRA ($r = 16$) shows an improvement of approximately 0.4% over LoRA ($r = 16$), while HydraLoRA ($r = 8$) demonstrates a 0.5% improvement. MELoRA ($r = 16$) shows no significant improvement. In contrast, TopLoRA ($r = 8$) uses fewer parameters while yielding significantly higher accuracy, highlighting the effectiveness of token-wise input-output projections. Moreover, increasing the rank of TopLoRA further enhances accuracy. This scalability will be discussed in more detail in Section 4.5.

## 4.3   Natural Language Generation Tasks

**Implementation Details.**   Both mathematical and commonsense reasoning benchmarks include a training corpus and multiple test sub-tasks. For each benchmark, we fine-tune the models on the training data and assess their performance across all sub-tasks. The learning rate is set to 1e-4 with 100 warm-up steps and linear decay, and the model is trained for one epoch. Both TopLoRA and baseline methods are applied to the query, key, and value weights. The implementation of these two reasoning tasks strictly adheres to the code provided in [11].

**Results on the Mathematical Reasoning Tasks.**   As shown in Table 2, TopLoRA achieves the highest accuracy in mathematical reasoning tasks. Compared to LoRA with the same rank ($r = 8$), TopLoRA improves the average accuracy of the three models by 1.67%, 3.65%, and 1.43%, respectively. Notably, these gains surpass those achieved by increasing LoRA's rank by a factor of four, demonstrating that TopLoRA's adaptive input-output projection significantly enhances LoRA's expressiveness. Regarding the three LoRA variants, MELoRA's performance is comparable to or even exceeds that of DoRA and HydraLoRA, unlike its results on the GLUE dataset. However, the overall improvements provided by these methods remain considerably smaller than those of TopLoRA.

Table 2: The accuracy of different methods on **Mathematical Reasoning** tasks with various pretrained models. The highest average precision is bolded, and the second-highest is underlined.

| | | #Params | AddSub | MultiArith | SingleEq | GSM8K | AQuA | SVAMP | Avg |
|---|---|---|---|---|---|---|---|---|---|
| Gemma-7B | LoRA($r = 8$) | 4.82M | 87.59 | 90.33 | 89.76 | 56.10 | 29.13 | 75.70 | 71.44 |
| | LoRA($r = 16$) | 9.63M | 86.84 | 92.83 | 89.57 | 58.15 | 30.71 | 74.90 | 72.17 |
| | LoRA($r = 32$) | 19.3M | 86.58 | 91.50 | 91.93 | 58.45 | 32.28 | 75.50 | 72.71 |
| | DoRA($r = 16$) | 9.98M | 87.59 | 94.17 | 91.34 | 58.68 | 27.95 | 75.80 | 72.59 |
| | MELoRA($r = 16$) | 9.63M | 87.26 | 92.22 | 91.01 | 59.49 | 32.15 | 74.97 | 72.85 |
| | HydraLoRA($r = 8$) | 11.1M | 87.34 | 92.67 | 90.16 | 58.83 | 27.95 | 75.50 | 72.08 |
| | **TopLoRA**($r = 8$) | 6.88M | 87.85 | 94.78 | 91.80 | 58.43 | 30.97 | 74.87 | 73.11 |
| | **TopLoRA**($r = 16$) | 13.8M | 86.33 | 94.83 | 92.52 | 59.29 | 31.10 | 75.60 | **73.28** |
| LLama-3-8B | LoRA($r = 8$) | 4.72M | 82.28 | 87.06 | 91.60 | 55.65 | 24.02 | 68.53 | 68.19 |
| | LoRA($r = 16$) | 9.44M | 84.56 | 91.22 | 92.26 | 57.22 | 25.72 | 70.17 | 70.19 |
| | LoRA($r = 32$) | 18.9M | 87.17 | 93.39 | 93.50 | 57.87 | 26.25 | 71.83 | 71.67 |
| | DoRA($r = 16$) | 9.63M | 85.95 | 89.67 | 92.62 | 56.52 | 26.19 | 70.40 | 70.23 |
| | MELoRA($r = 16$) | 9.44M | 85.82 | 87.83 | 91.54 | 55.34 | 24.41 | 71.20 | 69.36 |
| | HydraLoRA($r = 8$) | 9.04M | 86.08 | 91.00 | 91.14 | 55.50 | 25.98 | 68.10 | 69.63 |
| | **TopLoRA**($r = 8$) | 7.87M | 87.34 | 92.83 | 92.91 | 59.21 | 24.02 | 74.70 | 71.84 |
| | **TopLoRA**($r = 16$) | 15.7M | 88.86 | 92.17 | 93.31 | 61.11 | 28.74 | 73.50 | **72.95** |
| Qwen2.5-14B | LoRA($r = 8$) | 8.65M | 93.16 | 96.67 | 92.32 | 75.66 | 31.10 | 85.60 | 79.09 |
| | LoRA($r = 16$) | 17.3M | 91.90 | 96.33 | 92.91 | 74.37 | 34.65 | 86.40 | 79.43 |
| | LoRA($r = 32$) | 34.6M | 92.24 | 97.39 | 92.98 | 76.37 | 34.78 | 87.13 | 80.15 |
| | DoRA($r = 16$) | 17.6M | 92.41 | 96.39 | 92.39 | 75.92 | 36.22 | 86.80 | 80.02 |
| | MELoRA($r = 16$) | 17.3M | 92.91 | 97.33 | 92.13 | 75.89 | 33.86 | 85.60 | 79.62 |
| | HydraLoRA($r = 8$) | 16.4M | 92.41 | 96.22 | 92.45 | 76.32 | 36.61 | 86.97 | 80.16 |
| | **TopLoRA**($r = 8$) | 14.6M | 91.31 | 97.67 | 93.44 | 77.31 | 35.96 | 87.43 | 80.52 |
| | **TopLoRA**($r = 16$) | 29.1M | 91.65 | 98.50 | 93.90 | 75.74 | 37.40 | 87.40 | **80.76** |

**Results on the Commonsense Reasoning Tasks.** As shown in Table 3, TopLoRA also achieves the highest accuracy in commonsense reasoning tasks, and the experimental conclusions were consistent with those in mathematical reasoning tasks. At the same rank ($r = 8$), TopLoRA outperforms LoRA by 2.02%, 1.14%, and 0.61% in the average accuracy across the three models, respectively. In comparison, increasing LoRA's rank by four times yields improvements of 0.66%, 1.27%, and 0.45%. This demonstrates that TopLoRA can achieve comparable fine-tuning performance while reducing parameter requirements by nearly fourfold or more. Note that Qwen2.5-14B achieves very high accuracy in this task, making the improvements from various methods less noticeable. Additionally, TopLoRA outperforms the other three LoRA variants in accuracy while using fewer parameters.

## 4.4 Ablation Studies

To optimize the token-wise diagonal matrix $\Sigma_X$ effectively, TopLoRA incorporates RMSNorm and exponential functions into the projector $\Gamma_\Theta$, as shown in formula (7). To assess the impact of these components, we conducted an ablation study on mathematical reasoning tasks, fixing the rank at 8. The results in Table 4 indicate that removing either the exponential function or the normalization step significantly reduces model accuracy. However, omitting the normalization step results in a more substantial performance decline. This is due to the small values of $\Theta X$; without normalization, the elements in $\Sigma_X$ remain close to 1, limiting TopLoRA's ability to capture complex patterns. It is worth noting that TopLoRA reverts to standard LoRA when $\Sigma_X = I$.

## 4.5 Scalability Analysis

In this section, we evaluate the scalability of TopLoRA by varying the LoRA rank and tuning granularity. Our experiments focus on mathematical reasoning tasks using the LLama-3-8B model. To investigate the effect of different LoRA ranks, we tune the rank from the set $\{2, 4, 8, 16, 32, 64, 128\}$. The results, shown in Figure 2(a), demonstrate that TopLoRA consistently surpasses LoRA across all tested ranks. Next, we examine how TopLoRA scales when different tuning granularity are applied. By default, LoRA and TopLoRA are only applied to the query, key, and value weights. Here, we extend the evaluation to include four additional configurations: Q, QV, QKVUD, and

Table 3: The accuracy of different methods on **Commonsense Reasoning** tasks with various pre-trained models. The highest average precision is bolded, and the second-highest is underlined.

| | | #Param | BoolQ | PIQA | SIQA | HellaSwag | WinoGrande | ARC-c | ARC-e | OBQA | Avg |
|---|---|---|---|---|---|---|---|---|---|---|---|
| Gemma-7B | LoRA($r=8$) | 4.82M | 70.15 | 88.96 | 78.05 | 94.08 | 89.82 | 83.96 | 94.02 | 88.60 | 85.95 |
| | LoRA($r=16$) | 9.63M | 75.17 | 88.74 | 77.58 | 95.21 | 89.11 | 84.73 | 92.93 | 88.00 | 86.43 |
| | LoRA($r=32$) | 19.3M | 74.34 | 89.72 | 78.10 | 95.77 | 88.95 | 84.73 | 94.11 | 87.20 | 86.61 |
| | DoRA($r=16$) | 9.98M | 73.49 | 90.44 | 79.19 | 95.03 | 90.00 | 84.73 | 93.79 | 88.67 | 86.92 |
| | MELoRA($r=16$) | 9.63M | 73.49 | 89.50 | 79.99 | 94.60 | 89.90 | 84.30 | 93.18 | 89.40 | 86.79 |
| | HydraLoRA($r=8$) | 11.1M | 72.14 | 89.21 | 81.30 | 95.11 | 89.45 | 85.32 | 94.56 | 89.00 | 87.01 |
| | **TopLoRA($r=8$)** | 6.88M | 75.20 | 90.15 | 82.80 | 95.94 | 90.92 | 85.67 | 95.12 | 88.00 | 87.97 |
| | **TopLoRA($r=16$)** | 13.8M | 74.98 | 90.81 | 83.01 | 95.86 | 89.98 | 86.60 | 95.24 | 91.00 | **88.44** |
| LLama-3-8B | LoRA($r=8$) | 4.72M | 73.17 | 89.34 | 80.64 | 93.22 | 87.42 | 80.20 | 92.51 | 87.13 | 85.45 |
| | LoRA($r=16$) | 9.44M | 73.54 | 89.50 | 81.18 | 94.18 | 88.00 | 81.11 | 93.15 | 88.53 | 86.15 |
| | LoRA($r=32$) | 18.9M | 73.87 | 90.01 | 82.09 | 94.95 | 88.37 | 82.20 | 93.57 | 88.73 | 86.72 |
| | DoRA($r=16$) | 9.63M | 73.85 | 88.79 | 81.83 | 94.35 | 89.11 | 81.48 | 93.56 | 88.20 | 86.40 |
| | MELoRA($r=16$) | 9.44M | 73.59 | 89.77 | 81.85 | 94.84 | 88.29 | 82.00 | 93.06 | 88.93 | 86.54 |
| | HydraLoRA($r=8$) | 9.04M | 72.66 | 89.39 | 80.96 | 93.84 | 87.53 | 81.14 | 92.85 | 88.20 | 85.82 |
| | **TopLoRA($r=8$)** | 7.87M | 73.52 | 89.50 | 82.09 | 94.42 | 87.92 | 81.74 | 93.73 | 89.80 | 86.59 |
| | **TopLoRA($r=16$)** | 15.7M | 74.28 | 90.10 | 83.42 | 94.47 | 88.00 | 82.00 | 94.02 | 88.40 | **86.84** |
| Qwen2.5-14B | LoRA($r=8$) | 8.65M | 75.99 | 93.80 | 84.34 | 96.39 | 92.34 | 94.03 | 98.06 | 95.00 | 91.24 |
| | LoRA($r=16$) | 17.3M | 76.15 | 93.74 | 84.44 | 96.87 | 92.50 | 94.20 | 98.36 | 95.80 | 91.51 |
| | LoRA($r=32$) | 34.6M | 76.70 | 93.91 | 84.90 | 96.91 | 92.42 | 94.62 | 98.23 | 95.80 | 91.69 |
| | DoRA($r=16$) | 17.6M | 76.68 | 93.76 | 84.72 | 96.97 | 92.45 | 94.48 | 98.20 | 95.67 | 91.62 |
| | MELoRA($r=16$) | 17.3M | 76.97 | 93.89 | 84.90 | 97.12 | 91.95 | 94.62 | 98.06 | 96.33 | 91.73 |
| | HydraLoRA($r=8$) | 16.4M | 76.70 | 93.42 | 84.54 | 96.74 | 92.74 | 94.11 | 98.11 | 95.80 | 91.52 |
| | **TopLoRA($r=8$)** | 14.6M | 77.09 | 93.63 | 84.80 | 97.11 | 93.29 | 93.94 | 98.36 | 96.60 | 91.85 |
| | **TopLoRA($r=16$)** | 29.1M | 77.28 | 93.53 | 85.11 | 97.17 | 93.29 | 94.62 | 98.44 | 96.00 | **91.93** |

Table 4: The accuracy of TopLoRA on mathematical reasoning tasks using LLama-3-8B, without employing the exponential function or the RMSNorm operation.

| | | AddSub | MultiArith | SingleEq | GSM8K | AQuA | SVAMP | Avg |
|---|---|---|---|---|---|---|---|---|
| Gemma-7B | TopLoRA | 87.85 | 94.78 | 91.80 | 58.43 | 30.97 | 74.87 | **73.11** |
| | TopLoRA w.o. Exp | 88.10 | 93.67 | 90.94 | 58.15 | 31.10 | 75.10 | 72.84 |
| | TopLoRA w.o. RMSNorm | 88.10 | 92.83 | 90.94 | 58.23 | 29.53 | 76.50 | 72.69 |
| LLama-3-8B | TopLoRA | 87.34 | 92.83 | 92.91 | 59.21 | 24.02 | 74.70 | **71.84** |
| | TopLoRA w.o. Exp | 83.80 | 93.83 | 92.32 | 56.86 | 26.77 | 69.80 | 70.56 |
| | TopLoRA w.o. RMSNorm | 88.35 | 94.33 | 92.52 | 58.23 | 25.20 | 72.00 | 71.77 |
| Qwen2.5-14B | TopLoRA | 91.31 | 97.67 | 93.44 | 77.31 | 35.96 | 87.43 | **80.52** |
| | TopLoRA w.o. Exp | 92.24 | 96.61 | 92.85 | 76.22 | 35.83 | 86.13 | 79.98 |
| | TopLoRA w.o. RMSNorm | 90.38 | 96.17 | 92.52 | 75.74 | 38.98 | 87.60 | 80.23 |

QKVOGUD, where the symbols O, G, U, and D represent output, gate, up, and down projection weights, respectively. Under the same rank of 8, the results presented in Figure 2(b) further confirm that TopLoRA outperforms LoRA under various tuning granularity.

## 5 Related Work

To reduce fine-tuning overhead, LoRA [10] decomposes the weight update $\Delta W \in \mathbb{R}^{m \times n}$ into two low-rank matrices, $A \in \mathbb{R}^{r \times n}$ and $B \in \mathbb{R}^{m \times r}$, where the LoRA rank $r$ determines the number of trainable parameters. LoRA can be integrated into a model without altering its architecture or increasing inference overhead. Recent studies have investigated various aspects of LoRA to enhance its performance, including better optimization [27], initialization methods [31, 45, 18], learning rates [6], and dynamic parameter allocation [50, 22], as well as the use of higher ranks [35, 12, 4, 19]. For instance, DoRA [27] decomposes pretrained weights into magnitude and direction components. The magnitude is learned via a trainable vector, while the direction is updated using LoRA. PiSSA [31] and LoRA-GA [45] perform singular value decomposition (SVD) on pretrained weights and sampled gradients to initialize the matrices $A$ and $B$. Li et al. [18] present a comprehensive analysis of LoRA initialization, showing that non-zero initialization improves the robustness of fine-tuning to variations in learning rate. LoRA+ [6] uses a larger learning rate for the matrix $B$ to enhance fine-tuning performance. AdaLoRA [50] dynamically adjusts the LoRA rank for different layers during fine-tuning. VB-LoRA [22] combines low-rank matrices from different LoRA layers using a shared

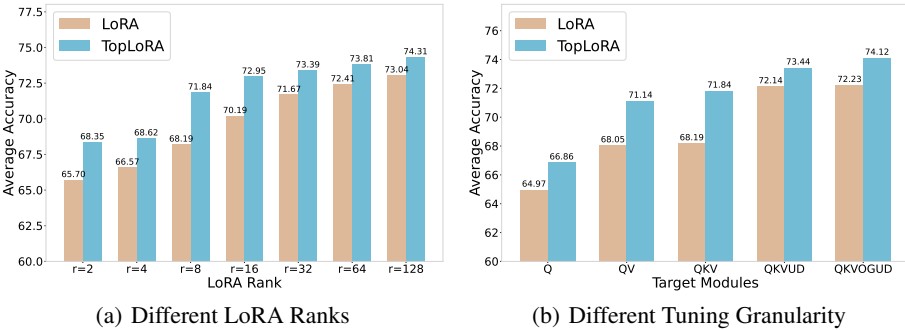

|     |     |
| --- | --- |
| (a) Different LoRA Ranks | (b) Different Tuning Granularity |

Figure 2: The scalability analysis on mathematical reasoning tasks using LLama-3-8B. (a) Accuracy of TopLoRA at varying ranks. (b) Accuracy of TopLoRA at different tuning granularity. The figures present only the average accuracy, and the detailed results are available in Appendix B.

vector bank. HiRA [12] and KronA [4] employ the Hadamard and Kronecker products, respectively, to enhance the rank of LoRA weights. Similar low-rank decompositions based on Hadamard and Kronecker products have also been widely used for efficient communication in distributed training [20, 46, 13]. MELoRA [35] achieves a higher rank by stacking low-rank matrices along the diagonal. ReLoRA [26] periodically merges learned LoRA adapters into the pretrained weights to increase the rank of weight updates. BoRA [19] reformulates LoRA as a block matrix multiplication, in which the low-rank factors are partitioned into multiple blocks and each block pair is modulated by a distinct diagonal matrix. This design enhances the diversity and rank of LoRA weights, substantially improving expressivity with minimal parameter overhead. More related work on LoRA variants and extensions is provided in Appendix C.

In this paper, we analyze LoRA from the perspective of input-output projections and express the LoRA weights as $BA = Q_B P Q_A$, where $Q_A$ and $Q_B$ denote the input and output spaces, and $P$ denotes the input-output projection. LoRA-XS [1] and LoRA-SB [32] demonstrate a similar structure, where the LoRA weight is decomposed as $BRA$, with $A \in \mathbb{R}^{r \times n}$ and $B \in \mathbb{R}^{m \times r}$ being frozen, while only $R \in \mathbb{R}^{r \times r}$ is trained. Essentially, these methods fix the input and output spaces, learning only the input-output projection. Different from existing studies, TopLoRA learns distinct input-output projections for different tokens to further improve the expressiveness of LoRA. Although the MoELoRA [30] and HydraLoRA [37] architectures use different weights for different tokens, they do not adequately address the limitations of shared input-output projections. Moreover, experimental results show that their performance does not match that of TopLoRA.

## 6 Conclusion

In this paper, we investigate LoRA from the perspective of input-output projections, decomposing the LoRA weights into three components: the input space, the output space, and the input-output projections. Beyond the LoRA rank, we identify another factor that limits its expressiveness: the shared input-output projection in LoRA is insufficient for capturing token-specific information. To address this limitation, we propose TopLoRA, which learns token-wise input-output projections in an end-to-end manner. Specifically, by adjusting the LoRA weights with token-wise diagonal matrices, TopLoRA enables finer-grained adaptation, even with a limited rank. Extensive experiments show that TopLoRA outperforms LoRA and its variants across a range of tasks and model scales. Notably, TopLoRA achieves a 2-4% accuracy improvement over LoRA at the same rank.

## Acknowledgements

This work is supported by the National Natural Science Foundation of China under grants 62376103, 62302184, 62436003 and 62206102; Major Science and Technology Project of Hubei Province under grant 2025BAB011 and 2024BAA008; Hubei Science and Technology Talent Service Project under grant 2024DJC078; and Ant Group through CCF-Ant Research Fund. The computation is completed in the HPC Platform of Huazhong University of Science and Technology.

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

# A  Detailed Experimental Settings

**Datasets and Models.** The GLUE benchmark [42] includes two single-sentence classification tasks (CoLA, SST-2), five pairwise text classification tasks (MNLI, RTE, QQP, MRPC, and QNLI), and one text similarity prediction task (STS-B). This paper reports the overall matched and mismatched accuracy for MNLI, Matthew's correlation for CoLA, Pearson correlation for STS-B, and accuracy for the remaining tasks. Due to the differing dataset sizes, the number of epochs varies: 10 epochs for RTE and MRPC, 5 epochs for STS-B and CoLA, 2 epochs for SST-2 and QNLI, and 1 epoch for MNLI and QQP. The models used are RoBERTa-Base and RoBERTa-Large [28].

**LoRA Hyperparameters.** The scaling factor is set to $\alpha = 2r$, where $r$ is the LoRA rank. LoRA is applied to the query and value weights with a dropout rate of 0.05, using full precision (FP32).

**Training Hyperparameters.** AdamW [29] is used with $\beta_1 = 0.9$, $\beta_2 = 0.999$, $\epsilon = 1e{-}8$, and no weight decay. The learning rate is selected from the set $\{3e{-}5, 1e{-}4, 3e{-}4, 1e{-}3\}$, with optimal values of $3e{-}4$ for RoBERTa-Base and $1e{-}4$ for RoBERTa-Large. A warm-up ratio of 0.03 is applied, and the batch size is set to 32. The maximum sequence length is 512.

## A.1  Experiments on Mathematical and Commonsense Reasoning Tasks

**Datasets.** Mathematical and commonsense reasoning tasks contain 10K and 170K training samples, respectively, along with several test tasks. Note that we directly utilize the data from [11] for our experiments. The training process consists of a single epoch. Three models are employed: Gemma-7B [36], LLama-3-8B [5], and Qwen2.5-14B [47].

**LoRA Hyperparameters.** The scaling factor is set to $\alpha = 2r$, where $r$ is the LoRA rank. LoRA is applied to the query, key, and value weights with a dropout rate of 0.05, using half precision (BF16).

**Training Hyperparameters.** AdamW is employed with the same settings as previously mentioned. The learning rate is chosen from the set $\{3e{-}5, 1e{-}4, 3e{-}4, 1e{-}3\}$ and is set to $1e{-}4$. A warm-up of 100 steps is applied, and the batch size is set to 16. The maximum sequence length is 256.

# B  Additional Experimental Results

## B.1  Standard Deviations

In the main experiments, each setting was repeated three times, and the average results are reported. For conciseness, standard deviations are provided in the Appendix. Table 5 shows the standard deviations for each GLUE dataset, where a separate model was trained for each. Table 6 presents the standard deviation of the average accuracy for the commonsense and mathematical reasoning tasks, where a single model was used across these sub-tasks. Notably, the standard deviation remains stable and is much smaller than the accuracy improvement achieved by TopLoRA.

Table 5: The standard deviation of different methods on the GLUE benchmark.

|  |  | RTE | MRPC | STS-B | CoLA | SST-2 | QNLI | MNLI | QQP |
|---|---|---|---|---|---|---|---|---|---|
| RoBERTa-Base | LoRA($r = 8$) | 0.51 | 0.46 | 0.27 | 0.57 | 0.11 | 0.15 | 0.11 | 0.10 |
|  | LoRA($r = 16$) | 0.84 | 0.40 | 0.34 | 0.64 | 0.29 | 0.11 | 0.04 | 0.08 |
|  | LoRA($r = 32$) | 0.78 | 0.35 | 0.39 | 0.56 | 0.14 | 1.00 | 0.11 | 0.03 |
|  | DoRA($r = 16$) | 0.95 | 0.46 | 0.31 | 0.57 | 0.14 | 0.02 | 0.24 | 0.11 |
|  | MELoRA($r = 16$) | 0.96 | 0.64 | 0.62 | 0.52 | 0.29 | 0.14 | 0.14 | 0.03 |
|  | HydraLoRA($r = 8$) | 0.59 | 0.87 | 0.75 | 0.61 | 0.34 | 0.17 | 0.21 | 0.08 |
|  | TopLoRA($r = 8$) | 0.89 | 0.31 | 0.39 | 0.38 | 0.61 | 0.14 | 0.11 | 0.03 |
|  | TopLoRA($r = 16$) | 0.72 | 0.58 | 1.01 | 0.47 | 0.29 | 0.03 | 0.23 | 0.16 |
| RoBERTa-Large | LoRA($r = 8$) | 0.55 | 0.81 | 0.26 | 0.64 | 0.50 | 0.22 | 0.23 | 0.05 |
|  | LoRA($r = 16$) | 0.51 | 0.20 | 0.59 | 0.98 | 0.19 | 0.16 | 0.27 | 0.04 |
|  | LoRA($r = 32$) | 0.40 | 0.66 | 0.13 | 0.65 | 0.33 | 0.44 | 0.32 | 0.02 |
|  | DoRA($r = 16$) | 0.48 | 0.20 | 0.13 | 0.56 | 0.14 | 0.33 | 0.13 | 0.07 |
|  | MELoRA($r = 16$) | 0.45 | 0.35 | 0.55 | 0.64 | 0.34 | 0.13 | 0.25 | 0.04 |
|  | HydraLoRA($r = 8$) | 0.67 | 0.72 | 0.21 | 0.29 | 0.09 | 0.09 | 0.40 | 0.09 |
|  | TopLoRA($r = 8$) | 0.90 | 0.31 | 0.07 | 0.16 | 0.09 | 0.35 | 0.15 | 0.04 |
|  | TopLoRA($r = 16$) | 0.34 | 0.58 | 0.14 | 0.78 | 0.14 | 0.12 | 0.25 | 0.10 |

Table 6: The standard deviation of the average accuracy on mathematical and commonsense reasoning tasks.

| | Mathematical Reasoning | | | Commonsense Reasoning | | |
|---|---|---|---|---|---|---|
| | Gemma-7B | LLama-3-8B | Qwen2.5-14B | Gemma-7B | LLama-3-8B | Qwen2.5-14B |
| LoRA($r = 8$) | 0.19 | 0.88 | 0.60 | 0.63 | 0.13 | 0.13 |
| LoRA($r = 16$) | 0.61 | 0.62 | 0.50 | 0.22 | 0.19 | 0.10 |
| LoRA($r = 32$) | 0.48 | 0.80 | 0.78 | 0.18 | 0.29 | 0.06 |
| DoRA($r = 16$) | 0.58 | 0.25 | 0.31 | 0.46 | 0.05 | 0.07 |
| MELoRA($r = 16$) | 0.31 | 0.67 | 0.32 | 0.56 | 0.16 | 0.18 |
| HydraLoRA($r = 8$) | 0.63 | 0.39 | 0.12 | 0.33 | 0.24 | 0.09 |
| TopLoRA($r = 8$) | 0.33 | 0.30 | 0.28 | 0.79 | 0.26 | 0.09 |
| TopLoRA($r = 16$) | 0.52 | 0.46 | 0.38 | 0.43 | 0.37 | 0.04 |

## B.2 Detailed Results of the Scalability Analysis

In Section 4.5, the scalability of TopLoRA was evaluated using LLama-3-8B on mathematical reasoning tasks from two perspectives: the LoRA rank and tuning granularity. Figure 2 presents the average accuracy of different settings across sub-tasks. Complete experimental results are available in Tables 7 and 8 for further comparison.

Table 7: The accuracy of LoRA and TopLoRA with varying target modules on mathematical reasoning tasks using LLama-3-8B.

| Target Modules | Method | #Params | AddSub | MultiArith | SingleEq | GSM8K | AQuA | SVAMP | Avg |
|---|---|---|---|---|---|---|---|---|---|
| Q | LoRA | 2.10M | 74.43 | 86.39 | 88.45 | 52.26 | 26.64 | 61.67 | 64.97 |
| | TopLoRA | 3.15M | 77.81 | 90.33 | 89.24 | 55.60 | 25.46 | 62.73 | 66.86 |
| QK | LoRA | 3.41M | 81.27 | 90.00 | 91.54 | 56.25 | 22.83 | 66.40 | 68.05 |
| | TopLoRA | 5.51M | 86.58 | 92.28 | 93.31 | 57.57 | 25.33 | 71.80 | 71.14 |
| QKV | LoRA | 4.72M | 82.28 | 87.06 | 91.60 | 55.65 | 24.02 | 68.53 | 68.19 |
| | TopLoRA | 7.87M | 87.34 | 92.83 | 92.91 | 59.21 | 24.02 | 74.70 | 71.84 |
| QKVUD | LoRA | 14.2M | 88.69 | 92.11 | 93.83 | 59.79 | 24.93 | 73.50 | 72.14 |
| | TopLoRA | 22.0M | 90.38 | 93.00 | 94.29 | 60.50 | 26.38 | 76.10 | 73.44 |
| QKVOGUD | LoRA | 21.0M | 87.59 | 93.28 | 93.90 | 59.41 | 24.15 | 75.03 | 72.23 |
| | TopLoRA | 30.9M | 89.28 | 96.33 | 94.69 | 61.41 | 27.17 | 75.87 | 74.12 |

Table 8: The accuracy of LoRA and TopLoRA with varying ranks on mathematical reasoning tasks using LLama-3-8B.

| Rank | Method | #Params | AddSub | MultiArith | SingleEq | GSM8K | AQuA | SVAMP | Avg |
|---|---|---|---|---|---|---|---|---|---|
| $r = 2$ | LoRA | 1.18M | 77.64 | 85.56 | 89.24 | 52.87 | 23.36 | 65.53 | 65.70 |
| | TopLoRA | 1.97M | 81.77 | 88.39 | 90.55 | 55.60 | 25.59 | 68.20 | 68.35 |
| $r = 4$ | LoRA | 2.36M | 79.49 | 86.83 | 88.85 | 55.12 | 23.10 | 66.03 | 66.57 |
| | TopLoRA | 3.93M | 82.11 | 89.61 | 91.27 | 56.94 | 25.20 | 66.57 | 68.62 |
| $r = 8$ | LoRA | 4.72M | 82.28 | 87.06 | 91.60 | 55.65 | 24.02 | 68.53 | 68.19 |
| | TopLoRA | 7.87M | 87.34 | 92.83 | 92.91 | 59.21 | 24.02 | 74.70 | 71.84 |
| $r = 16$ | LoRA | 9.44M | 84.56 | 91.22 | 92.26 | 57.22 | 25.72 | 70.17 | 70.19 |
| | TopLoRA | 15.7M | 88.86 | 92.17 | 93.31 | 61.11 | 28.74 | 73.50 | 72.95 |
| $r = 32$ | LoRA | 18.9M | 87.17 | 93.39 | 93.50 | 57.87 | 26.25 | 71.83 | 71.67 |
| | TopLoRA | 31.5M | 89.62 | 93.67 | 94.29 | 60.42 | 27.17 | 75.20 | 73.39 |
| $r = 64$ | LoRA | 37.7M | 88.10 | 94.33 | 93.70 | 60.05 | 25.20 | 73.10 | 72.41 |
| | TopLoRA | 62.9M | 90.89 | 97.17 | 94.88 | 60.42 | 24.41 | 75.10 | 73.81 |
| $r = 128$ | LoRA | 75.5M | 89.87 | 94.28 | 93.04 | 61.13 | 24.54 | 75.40 | 73.04 |
| | TopLoRA | 125.8M | 90.38 | 96.00 | 94.88 | 61.56 | 28.74 | 74.30 | 74.31 |

# C   More Related Work

In Section 5, we discuss various LoRA variants. In addition to LoRA, two other widely used PEFT methods are the adapter-based and soft prompt-based approaches. The adapter-based method [9, 7, 39] introduces new layers into the model and fine-tunes only these layers, significantly reducing resource consumption. The soft prompt-based method [14, 21, 34] adds learnable soft tokens (prompts) to the input, enabling the model to adapt to specific tasks. Through effectiveness, these methods typically introduce computational overhead during inference and thus increase inference latency. In contrast, LoRA applies weight updates directly to the pretrained weights after fine-tuning, avoiding additional inference latency.

More broadly, LoRA can be regarded as a compression technique. A variety of model compression methods have been proposed to reduce computational and storage costs, such as quantization [15], pruning [16], and knowledge distillation [43]. We expect to see tighter integration between LoRA and these complementary approaches to further enhance efficiency. Beyond efficiency, it is also valuable to explore the deployment of LoRA and TopLoRA in distributed environments [24, 44, 17] and continual learning scenarios [25, 23], as well as in addressing out-of-distribution generalization under distributional shifts [33], which represent promising directions for future research.

# D   Limitations

A potential limitation of TopLoRA is the inference latency. TopLoRA shows improved accuracy over standard LoRA; however, it relies on token-wise LoRA weights, which cannot be directly integrated into pretrained weights after fine-tuning. This requires additional computations during inference and thus increases inference latency. Similar trade-offs between dynamic adaptation and computational overhead are observed in related approaches, such as MoELoRA [30] and HydraLoRA [37]. Future work could explore optimizations to reduce inference costs while preserving TopLoRA's superiority. In addition, this paper does not address certain aspects of TopLoRA: (1) Beyond language models, can TopLoRA be applied to other (e.g., visual) tasks? (2) Can more theoretical or convincing explanations be provided to justify the advantages of TopLoRA? We are actively investigating these questions. Nonetheless, we are encouraged by the promising results of TopLoRA in our current experiments and look forward to further tests and feedback from the community.

# E   Broader Impacts

This paper presents TopLoRA, a method that enhances LoRA by learning token-wise input-output projections. TopLoRA achieves the same fine-tuning performance as LoRA while utilizing fewer parameters, thereby reducing computational overhead and promoting energy efficiency. Additionally, TopLoRA paves the way for new research directions in dynamic weight LoRA. As an extension of the established LoRA framework, TopLoRA does not introduce any significant social concerns that would require further discussion.

