# OpenReview forum: "Beyond Higher Rank: Token-wise Input-Output Projections for Efficient Low-Rank Adaptation"
_NeurIPS.cc/2025/Conference — NeurIPS 2025 poster_

### Official Review · Reviewer_2FJn · 2025-06-23

**Clarity:** 1
**Significance:** 2
**Originality:** 3
**Rating:** 4
**Confidence:** 4

**Summary:**

The author proposes a token-wise projected LoRA method, which dynamically adjusts the LoRA weights for different tokens. It outperforms the standard LoRA method across multiple benchmarks.

**Questions:**

Q1: While the paper spends a lot of effort discussing its motivations, it lacks experimental support for its claimed advantages. Can the authors provide additional explanations or empirical evidence to better justify the motivations behind their proposed approach?

Q2: In the abstract, the authors state that standard LoRA "all input tokens share the same weight", which seems to contradict the traditional LoRA framework. Can the authors explain further and how it differs from the TopLoRA design?

Q3: The paper TopLoRA claims to improve expressiveness under low-rank constraints, but TopLoRA introduces about 50% more parameters than LoRA of the same rank. How do the authors distinguish between the performance improvement brought by the architectural improvements and the increased number of parameters?

Q4: Since $\Delta W$ becomes token-dependent, does this method give up the mergeability and inference efficiency advantages that LoRA usually has? Can the authors quantify the additional inference cost and comment on whether this trade-off is reasonable?

Q5: Why are relevant baselines such as HiRA, KronA, and AdaLoRA omitted from the comparison? Inclusion of these approaches provides a clearer presentation of the relative strengths and weaknesses of TopLoRA.

**Ethical Concerns:**

["NO or VERY MINOR ethics concerns only"]

**Final Justification:**

The author provides a more in-depth motivation experiment and achieves better results compared with multiple baseline methods under fair conditions, which solves my concern, so I raise the score to 4.

**Limitations:**

Yes

**Paper Formatting Concerns:**

There are no major formatting issues identified in this paper.

**Quality:**

2

**Strengths And Weaknesses:**

Strengths:

The authors demonstrate superior results in multiple datasets.

Weaknesses:

1. The authors devote a substantial portion of the paper to describing the motivation, but lack experimental support.
2. The authors mention in the abstract that "all input tokens share the same weights." This appears to contradict the standard LoRA design, where each module has its own set of independent weights without weight sharing across tokens. A more detailed explanation or clarification would be appreciated.
3. While the authors state in line 116 that TopLoRA improves the expressiveness of LoRA under low-rank settings, the experiments reveal that, for the same rank, TopLoRA introduces about 50% more parameters than standard LoRA. This raises the question of whether the observed performance improvements are due to the proposed design or merely the increased number of parameters. The current experiments lack sufficient evidence to disentangle these two factors.
4. Since the proposed method learns a distinct $\theta$ for each token, $\Delta W$ becomes token-dependent. This raises a concern about whether the approach sacrifices the mergeability advantage of standard LoRA and incurs additional computational cost during inference. It would be helpful if the authors could clarify this trade-off and discuss its impact on inference efficiency.
5. The authors have not compared their method with other rank enhancement techniques such as HiRA and KronA, or with rank adaptation methods like AdaLoRA. Including such comparisons would strengthen the experimental evaluation and better position the proposed approach within the existing literature.
6. This paper lacks a related work section.
7. There is an error in line 40: a unitary matrix must be square, but $Q_B$ and $Q_A$ do not satisfy this condition. This needs to be corrected or further clarified.

---

> ### Author Rebuttal · Authors · 2025-07-30
>
> **Hi, Reviewer 2FJn:**
>
> Thank you for recognizing TopLoRA’s **superior accuracy**, and for your detailed and insightful feedback. It appears that your main concerns relate to the **empirical support for our motivation** and how our method **compares to other baseline methods**. Please find our point-by-point responses below.
>
> ***`Q1: "lack experimental support for the motivation."`***
>
> **R1:**
> As noted in Eq.(1) of our paper, the standard LoRA formulation $BA=Q_BPQ_A$projects token representations into a fixed low-dimensional subspace spanned by $Q_A$. This inherently limits the model’s ability to capture token-specific variations: any input component orthogonal to this subspace is irrecoverably lost. Consequently, when the rank is small—as is often necessary for efficiency—the amount of discarded token-specific information increases.
>
> To further quantify this limitation, we compute the **mutual information (MI)** between input tokens $X$ and output representations $Y$ using the InfoNCE lower bound [1], a standard technique in representation learning. Each input-output pair $(X_i,Y_i)$ is treated as positive, with others as negatives. A higher MI indicates better preservation of token-specific information. This method allows us to provide a tractable, quantitative comparison of how much token-level information is retained.
>
> Due to the image upload limitations during rebuttal, we summarize a subset of the results on LLaMA-3-8B (Math) in the table below and will include full visualizations and analysis in the revised paper.
>
> ||Layer5|Layer10|Layer20|
> |-|-|-|-|
> |LoRA($r=8$)|1.13|0.62|1.96|
> |LoRA($r=16$)|1.56|0.76|2.12|
> |TopLoRA($r=8$)|2.31|1.02|2.97|
>
> These results show that **TopLoRA retains significantly more mutual information than standard LoRA**, even at smaller ranks, confirming its stronger ability to capture token-specific features. Together with the consistent performance improvements shown in our experiments, this provides strong support for our central claim: TopLoRA’s dynamic modulation enhances the expressiveness of LoRA.
>
> [1] A. van den Oord et al., Representation Learning with Contrastive Predictive Coding, 2018.
>
>
> ***`Q2: "each (LoRA) module has its own set of independent weights without weight sharing across tokens."`***
>
> **R2:**
> We believe there may be a misunderstanding. When we state that different tokens share the same LoRA weights, we mean that **within a given LoRA layer (or module), each input token $X$ (from a sequence of multiple tokens) is processed using the same set of LoRA weights**. This can be formally expressed as $Y=(W+BA)X$, where the same LoRA weights $BA$ are applied uniformly to all tokens in the sequence.
> To introduce token-specific modulation, we propose generating a diagonal matrix $\Sigma_X$ based on the input token $X$, which is used to adjust the LoRA weights for each token in the form $B\Sigma_X A$.
>
> ***`Q3: "This raises the question of whether the performance improvements are due to the proposed design or merely the increased number of parameters."`***
>
> **R3:**
> Due to its improved structure, TopLoRA introduces approximately 50% more parameters than LoRA at the same rank.
> However, the performance gains it achieves significantly surpass those obtained by merely increasing the rank of LoRA.
> As noted in line 227, "these gains surpass those achieved by increasing LoRA’s rank by a factor of four".
> For example, on Mathematical Reasoning tasks with Gemma-7B, LoRA (r=32) requires **19.3M** trainable parameters to achieve an accuracy of **72.71**, whereas TopLoRA (r=8) achieves a higher accuracy of **73.11** with only **6.88M** parameters.
>
> To ensure a fair comparison, we evaluated TopLoRA with ranks of 5, 10, and 20, and compared it with LoRA at ranks of 8, 16, and 32. In these settings, both methods use a similar number of parameters. As shown in the table below, we report the average accuracy of different models on mathematical and commonsense reasoning benchmarks. The *Accuracy Gain* column indicates the performance improvement of TopLoRA over LoRA under comparable parameter budgets.
> **TopLoRA achieves a 1–3% boost in accuracy with a similar number of parameters.** Notably, increasing the LoRA rank by a factor of four yields only a modest **1–2%** improvement.
> Therefore, we can confidently attribute TopLoRA’s performance gains to its structural design, rather than the approximately 50% increase in parameter count.
>
> |||#Params|Mathematical|Accuracy Gain|Commonsense|Accuracy Gain|
> |-|-|-|-|-|-|-|
> |Gemma-7B|LoRA($r=8$)|4.82M|71.44|-|85.95|-|
> ||TopLoRA($r=5$)|4.30M|72.97|**1.53**|87.68|**1.73**|
> ||LoRA($r=16$)|9.63M|72.17|-|86.43|-|
> ||TopLoRA($r=10$)|8.60M|73.16|**0.99**|87.92|**1.49**|
> ||LoRA($r=32$)|19.3M|72.71|-|86.61|-|
> ||TopLoRA($r=20$)|17.2M|74.14|**1.43**|88.44|**1.83**|
> |Llama-3-8B|LoRA($r=8$)|4.72M|68.19|-|85.45|-|
> ||TopLoRA($r=5$)|4.92M|70.94|**2.75**|86.37|**0.92**|
> ||LoRA($r=16$)|9.44M|70.19|-|86.15|-|
> ||TopLoRA($r=10$)|9.83M|72.37|**2.18**|86.68|**0.53**|
> ||LoRA($r=32$)|18.9M|71.67|-|86.72|-|
> ||TopLoRA($r=20$)|19.7M|73.52|**1.85**|87.92|**1.20**|
> |Qwen2.5-14B|LoRA($r=8$)|8.65M|79.09|-|91.24|-|
> ||TopLoRA($r=5$)|9.09M|80.11|**1.02**|91.60|**0.36**|
> ||LoRA($r=16$)|17.3M|79.43|-|91.51|-|
> ||TopLoRA($r=10$)|18.2M|80.87|**1.44**|91.93|**0.42**|
> ||LoRA($r=32$)|34.6M|80.15|-|91.69|-|
> ||TopLoRA($r=20$)|36.4M|81.25|**1.10**|92.85|**1.16**|
>
> On the other hand, inspired by Reviewer SWr9, we explored **sharing a common set of learnable projectors across different layers** for weight modulation. The results show that **a small number of shared projectors—approximately 8—is sufficient to achieve performance comparable to the original TopLoRA**. This strategy offers an effective means of further reducing the number of trainable parameters in TopLoRA without sacrificing accuracy. Please kindly check **Q5** in our response to Reviewer **SWr9** for experimental results.
>
>
> ***`Q4: "Additional computational cost during inference."`***
>
> **R4:**
> As described in Appendix C, we explicitly acknowledge this limitation: TopLoRA introduces token-dependent LoRA weights that cannot be merged back into the pretrained model after fine-tuning. As a result, this design incurs additional computational overhead during inference and may lead to increased latency.
>
> **Similar limitations are observed in methods such as MoELoRA and HydraLoRA** ([19,25] in our paper), which also adopt dynamic adaptation mechanisms to enhance expressiveness at the cost of higher inference complexity. Likewise, TopLoRA is best suited for scenarios where merging LoRA weights is not required; for example, in **multi-task settings** involving multiple LoRA modules.
>
> ***`Q5: "Comparison with HiRA, KronA and AdaLoRA."`***
>
> **R5:**
> As suggested, we added comparisons with HiRA, KronA, and AdaLoRA.
> HiRA and KronA enhance LoRA by using the Hadamard and Kronecker products, respectively, to increase the rank of LoRA weights. AdaLoRA, on the other hand, dynamically adjusts the LoRA rank across layers during fine-tuning.
>
> In the comparison, we primarily set the LoRA rank to 16. It is important to note that KronA does not involve a rank hyperparameter, and its number of trainable parameters is fixed and cannot be adjusted.
> Due to their structural modifications, HiRA and KronA require larger learning rates to perform well. Therefore, we tuned the learning rate among {1e-4, 3e-4, 1e-3, 3e-3, 1e-2} and report their best accuracy. AdaLoRA is configured with an initial rank of 24 and a target rank of 16.
> For TopLoRA, we used a rank of 10 to ensure comparable parameter counts with the baselines.
>
> As shown below, we report the average accuracy of different models on mathematical and commonsense reasoning benchmarks. The *Accuracy Gain* column reflects the performance improvement over standard LoRA.
> KronA, likely due to its extremely low parameter count, shows worse performance than LoRA. HiRA and AdaLoRA provide moderate improvements, but the gains are less pronounced compared to TopLoRA.
>
> It is also worth noting that TopLoRA's improvements vary across datasets and models. For instance, Qwen2.5-14B already achieves strong performance on commonsense reasoning tasks, so the additional gain from TopLoRA is relatively modest (0.42%). However, this still exceeds the improvement from doubling the LoRA rank, which only yields a 0.18% increase (as shown in the table in Q3).
>
>
> |||#Params|Mathematical|Accuracy Gain|Commonsense|Accuracy Gain|
> |-|-|-|-|-|-|-|
> |Gemma-7B|LoRA($r=16$)|9.63M|72.17|-|86.43|-|
> ||KronA|0.60M|69.46|**-2.71**|84.49|**-1.94**|
> ||HiRA($r=16$)|9.63M|72.91|**0.74**|87.09|**0.66**|
> ||AdaLoRA($r=16$)|9.63M|72.65|**0.48**|86.98|**0.55**|
> ||TopLoRA($r=10$)|8.60M|73.16|**0.99**|87.92|**1.49**|
> |Llama-3-8B|LoRA($r=16$)|9.44M|70.19|-|86.15|-|
> ||KronA|0.52M|67.41|**-2.78**|83.97|**-2.18**|
> ||HiRA($r=16$)|9.44M|70.96|**0.77**|86.38|**0.23**|
> ||AdaLoRA($r=16$)|9.44M|70.50|**0.31**|86.20|**0.05**|
> ||TopLoRA($r=10$)|9.83M|72.37|**2.18**|86.68|**0.53**|
> |Qwen2.5-14B|LoRA($r=16$)|17.3M|79.43|-|91.51|-|
> ||KronA|0.95M|78.73|**-0.70**|89.36|**-2.15**|
> ||HiRA($r=16$)|17.3M|79.59|**0.16**|91.78|**0.27**|
> ||AdaLoRA($r=16$)|17.3M|79.65|**0.22**|91.49|**-0.02**|
> ||TopLoRA($r=10$)|18.2M|80.87|**1.44**|91.93|**0.42**|
>
>
>
> ***`Q6: "This paper lacks a related work section."`***
>
> **R6:** The related work is currently presented in Section 5 to maintain a coherent narrative, allowing readers to first gain a complete understanding of our method before engaging with comparisons to prior approaches.
> We will carefully consider relocating the related work to Section 2 in the revised version to improve clarity and alignment with standard conventions.
>
>
> ***`Q7: "There is an error in line 40: a unitary matrix must be square."`***
>
> **R7:**
> Thank you for pointing out the error. The correct statement is that both $Q_A$ and $Q_B$ are **semi-unitary** matrices; that is, $Q_A Q_A^\top=I$ and $Q_B^\top Q_B=I$.

---

> > ### Comment · Reviewer_2FJn · 2025-08-04
> > **Thanks for your response.**
> >
> > Thanks for the clarification and response. My concerns have been resolved. I think adding motivation experiments will help improve the quality of the article. I raised my rating to 4.

---

> > > ### Author Response · Authors · 2025-08-04
> > > **Sincere Thanks for Your Feedback and Support**
> > >
> > > Thank you for your thoughtful review, constructive feedback, and for raising your score. We're glad our clarifications addressed your concerns. We greatly appreciate your suggestion regarding motivation experiments and will carefully revise the paper accordingly.

---

### Official Review · Reviewer_SWr9 · 2025-06-28

**Clarity:** 4
**Significance:** 3
**Originality:** 3
**Rating:** 5
**Confidence:** 4

**Summary:**

This paper reformulates the low-rank adapter to show the limited accessibility of the current LoRA residuals to input-output subspaces. Coupled with the static behavior of conventional LoRA implementation when it comes to different tokens in varying contexts, they propose to modify this formulation to allow for token-dependent adjustments on the residual during both training and inference. This is done by adding an additional set of weights that transform the activations from the previous layer using normalization and exponential operators to a new vector. This vector is then added to a diagonal matrix within the new formulation of the adapter.

**Questions:**

- Regarding the statement “different tokens exhibit distinct semantics and distributions, and the shared input-output projection” in L51, given that both model weights and LoRA weights can learn to map different tokens differently, the only way for me to make sense of this statement is to relate it to the rank again. That way, I can interpret this in the sense that when the rank is low, there are fewer subspaces available to express every concept represented by the tokens in the fine-tuning dataset and additional token-dependent mechanisms are required. Is that what you mean by L51? The reason for the question is that you start the paragraph with “beyond the LoRA rank…” and that makes me think that you’re stating this is something completely disentangled with respect to the rank.

**Ethical Concerns:**

["NO or VERY MINOR ethics concerns only"]

**Final Justification:**

All my concerns in the initial review were addressed by the authors during rebuttal so I changed my score to 5 for my final review. I don't necessarily agree with the reasons of rejection by Reviewer 2FJn, especially considering the rebuttal answers (the answer to Q3 of this reviewer being the most important one).

**Limitations:**

- An additional limitation would also be the dependence of the additional weights on the rank of the adapter. This also makes me wonder if it is possible to share the overhead between layers by attributing a single $\Sigma_X$ operation for the whole model.

**Quality:**

3

**Strengths And Weaknesses:**

Strengths:
- The writing is clear and fluid.
- The experiments are comprehensive and tests are done on a variety of models.
- The proposed reformulation is straightforward.

Weaknesses:
- The statement "LoRA typically exhibits a performance gap when compared to FFT" in L29 means that you'd need to compare to methods addressing this gap. You've already compared to DoRA but other methods that address the same problem are not compared against, e.g. LoRA+ [a].
- Similar works like MoELoRA have already been mentioned by the authors but dismissed due to differences in goals. I understand the difference in goals and that MoELoRA should not be compared against as a SoTA for this work but it would have been complementary to see combinations of the two works as an additional experiment.
- A parameter count on Figure 2 would complement the visualization.

[a] Hayou, Soufiane, Nikhil Ghosh, and Bin Yu. "LoRA+: Efficient Low Rank Adaptation of Large Models." Proceedings of the Forty-first International Conference on Machine Learning, 2024. https://openreview.net/forum?id=NEv8YqBROO.

---

> ### Author Rebuttal · Authors · 2025-07-30
>
> **Hi, Reviewer SWr9:**
>
> Thank you for your positive assessment of the **clarity** of our writing and the **comprehensiveness** of our experiments.
> Regarding your questions on **comparisons with LoRA+** and the **integration with MoELoRA**, we have conducted substantial additional analysis to strengthen our response. Please find our point-by-point responses below.
>
> ***`Q1: "Comparison with LoRA+."`***
>
> **R1:**
> Following your suggestion, we have added comparisons with LoRA+.
>
> The core idea of LoRA+ is to apply a larger learning rate to the $B$ matrix, i.e., $\beta = \eta_B/\eta_A >1$.
> We tuned $\beta$ over the range {2, 4, 8, 16}, and reported the best-performing result. The rank of both LoRA and LoRA+ is set to 16, while for TopLoRA, we used a rank of 10 to ensure comparable parameter counts across methods.
>
> As shown below, we report the average accuracy of different models on mathematical and commonsense reasoning benchmarks. The *Accuracy Gain* column reflects the performance improvement over standard LoRA. With a similar number of parameters, **TopLoRA consistently achieves higher accuracy than LoRA+**.
>
> Importantly, **our method is orthogonal to LoRA+**. Specifically, we can also apply a larger learning rate to the $B$ matrix in TopLoRA—a variant we refer to as **TopLoRA+**, which further improves accuracy by approximately 1%. These results will be included in the revised version of the paper.
>
> |  |  | #Params | Mathematical | Accuracy Gain | Commonsense | Accuracy Gain  |
> |---|---|---|---|---|---|---|
> | Gemma-7B | LoRA ($r=16$) | 9.63M | 72.17 | - | 86.43 | -  |
> |  | LoRA+ ($r=16$) | 9.63M | 72.74 | **0.57**  | 87.48 | **1.05**   |
> |  | TopLoRA ($r=10$) | 8.60M | 73.16 | **0.99**  | 87.92 | **1.49**   |
> |  | TopLoRA+ ($r=10$) | 8.60M | 73.86 | **1.69**  | 88.57 | **2.14**   |
> | Llama-3-8B | LoRA ($r=16$) | 9.44M | 70.19 | - | 86.15 | -  |
> |  | LoRA+ ($r=16$) | 9.44M | 71.61 | **1.42**  | 86.5 | **0.35**   |
> |  | TopLoRA ($r=10$) | 9.83M | 72.37 | **2.18**  | 86.68 | **0.53**   |
> |  | TopLoRA+ ($r=10$) | 9.83M | 73.59 | **3.40**  | 87.33 | **1.18**   |
> | Qwen2.5-14B | LoRA ($r=16$) | 17.3M | 79.43 | - | 91.51 | -  |
> |  | LoRA+ ($r=16$) | 17.3M | 80.39 | **0.96**  | 91.62 | **0.11**   |
> |  | TopLoRA ($r=10$) | 18.2M | 80.87 | **1.44**  | 91.93 | **0.42**   |
> |  | TopLoRA+ ($r=10$) | 18.2M | 81.34 | **1.91**  | 92.06 | **0.55**   |
>
>
> ***`Q2: "Combination of MoELoRA and TopLoRA."`***
>
> **R2:** Based on your suggestions, we conducted a more in-depth investigation of MoELoRA. First, we added experiments using MoELoRA with four fully activated experts, each implemented as a rank-4 LoRA module. This configuration results in a parameter count comparable to that of a rank-16 TopLoRA.
>
> As shown below, we report the average accuracy of different models on mathematical and commonsense reasoning benchmarks. The *Accuracy Gain* column reflects the performance improvement over LoRA ($r=16$).
> The results show that TopLoRA significantly outperforms MoELoRA under a similar parameter budget.
>
> Furthermore, we replaced each expert in MoELoRA with a rank-4 TopLoRA module, creating a variant we refer to as **MoETopLoRA**. Our findings indicate that the performance gains from MoELoRA and TopLoRA are largely additive, resulting in further improvements. The corresponding content will be included in the revised version of the paper.
>
> |  |  | #Params | Mathematical | Accuracy Gain | Commonsense | Accuracy Gain  |
> |---|---|---|---|---|---|---|
> | Gemma-7B | LoRA ($r=16$) | 9.63M | 72.17 | - | 86.43 | -  |
> |  | MoELoRA ($r=4,n=4$) | 10.7M | 72.3 | **0.13** | 86.80 | **0.37**  |
> |  | TopLoRA ($r=16$) | 13.8M | 73.28 | **1.11** | 87.45 | **1.02**  |
> |  | MoETopLoRA ($r=4,n=4$) | 14.9M | 73.49 | **1.32** | 87.62 | **1.19**  |
> |  | LoRA ($r=32$) | 19.3M | 72.71 | **0.54** | 86.61 | **0.18**  |
> | Llama-3-8B | LoRA ($r=16$) | 9.44M | 70.19 | - | 86.15 | -  |
> |  | MoELoRA ($r=4,n=4$) | 11.0M | 71.36 | **1.17** | 86.42 | **0.27**  |
> |  | TopLoRA ($r=16$) | 15.7M | 72.95 | **2.76** | 86.84 | **0.69**  |
> |  | MoETopLoRA ($r=4,n=4$) | 17.2M | 73.33 | **3.14** | 87.28 | **1.13**  |
> |  | LoRA ($r=32$) | 18.9M | 71.67 | **1.48** | 86.72 | **0.57**  |
> | Qwen2.5-14B | LoRA ($r=16$) | 17.3M | 79.43 | - | 91.51 | -  |
> |  | MoELoRA ($r=4,n=4$) | 20.3M | 79.83 | **0.40** | 91.44 | **-0.07**  |
> |  | TopLoRA ($r=16$) | 29.1M | 80.76 | **1.33** | 91.71 | **0.20**  |
> |  | MoETopLoRA ($r=4,n=4$) | 32.1M | 80.65 | **1.22** | 91.73 | **0.22**  |
> |  | LoRA ($r=32$) | 34.6M | 80.15 | **0.72** | 91.69 | **0.18**  |
>
>
> ***`Q3: "Parameter count on Figure 2."`***
>
> **R3:** Thank you for the suggestion. We will add an additional vertical axis in Figure 2 to indicate the number of parameters. As image updates are not allowed during the rebuttal stage, we are currently unable to present the revised figure. However, this is a relatively straightforward modification, and we will incorporate it in the revised version.
>
>
> ***`Q4: "the only way for me to make sense of this statement (L51) is to relate it to the rank again."`***
>
> **R4:** Below is our response to the question raised by the reviewer in the Questions section.
>
> First, we believe that the reviewer’s understanding is correct, our proposed method is not completely rank-independent. As shown in Equation (1), the LoRA weights are expressed as $𝐵𝐴=Q_B P Q_A$. The rank determines the dimensions of the input space ($Q_A$) and output space ($Q_B$); increasing the rank expands both, thereby enhancing the expressive capacity of LoRA.
>
> In this paper, we argue that beyond increasing the rank, LoRA’s expressiveness can also be improved by enriching the mapping from the input space to the output space. In standard LoRA, a fixed projection matrix $P$ is applied uniformly to all tokens $X$. We propose generating a diagonal matrix $\Sigma_X$ based on the input token $X$ to modulate $P$, thereby introducing token-wise adaptation and improving expressive capacity of LoRA.
>
> We realize that the reviewer’s question likely stems from the phrasing "beyond the LoRA rank..." in Line 51. A more precise statement would be "beyond increasing the LoRA rank..." which is consistent with our framing in the title: "Beyond Higher Rank".
>
>
> ***`Q5: "It is possible to share the overhead between layers by attributing a single operation for the whole model."`***
>
> **R5:**
> Thank you for this insightful suggestion, it was truly inspiring for us. Indeed, the learnable projector in TopLoRA can be reconsidered from the perspective of sharing across layers.
>
> In the original design of TopLoRA, each layer learns a distinct projector to modulate the LoRA weights based on the input token. However, if we temporarily ignore the layer-wise structure, input tokens from different layers can be grouped together and share a common set of projectors for weight modulation.
>
> Motivated by this idea, we conducted experiments on LLaMA-3-8B with a rank of 8, sharing a fixed number of learnable projectors $N\in\\{1,2,4,8,16\\}$ across different layers. As shown below, we report the average accuracy on mathematical and commonsense reasoning benchmarks. The *Accuracy Gain* column indicates the performance improvement relative to standard LoRA.
> The results show that **a small number of shared projectors—approximately 8—is sufficient to achieve performance comparable to the original TopLoRA**. This strategy offers an effective means of further reducing the number of trainable parameters in TopLoRA without sacrificing accuracy.
> The revised version of the paper will include these discussions and results.
>
> Once again, we sincerely thank the reviewer for this valuable comment, which has meaningfully improved our work.
>
> | LLaMA-3-8B ($r=8$) | #Params | Mathematical | Accuracy Gain | Commonsense | Accuracy Gain  |
> |---|---|---|---|---|---|
> | LoRA | 4.72M | 68.19 | - | 85.45 | -  |
> | TopLoRA | 7.87M | 71.84 | **3.65**  | 86.59 | **1.14**   |
> | TopLoRA ($N=1$) | 4.75M | 68.39 | **0.20**  | 85.56 | **0.11**   |
> | TopLoRA ($N=2$) | 4.78M | 69.06 | **0.87**  | 86.26 | **0.81**   |
> | TopLoRA ($N=4$) | 4.85M | 70.63 | **2.44**  | 86.17 | **0.72**   |
> | TopLoRA ($N=8$) | 4.98M | 71.98 | **3.79**  | 86.34 | **0.89**   |
> | TopLoRA ($N=16$) | 5.24M | 71.97 | **3.78**  | 86.55 | **1.10**  |

---

> > ### Comment · Reviewer_SWr9 · 2025-08-04
> >
> > Thank you for the clarifications and the extensive additional experiments. All my concerns have been addressed. I'm raising my score to 5.

---

> > > ### Author Response · Authors · 2025-08-04
> > > **Sincere Thanks for Your Feedback and Support**
> > >
> > > Thank you again for reviewing our paper and for your thoughtful and constructive feedback. We’re pleased to hear that our additional clarifications and experiments addressed your concerns. We sincerely appreciate your recognition of our efforts and your decision to raise the score. Your comments were highly valuable in improving the quality of this work.

---

### Official Review · Reviewer_6dXU · 2025-07-02

**Clarity:** 3
**Significance:** 2
**Originality:** 2
**Rating:** 4
**Confidence:** 3

**Summary:**

This paper proposes a PEFT method called Token-wise Projected Low-Rank Adaptation (TopLoRA), which dynamically adjusts LoRA weights according to the input token. Specifically, it learns a mapping that projects the input token to a scaling matrix that is further multiplied by the LoRA matrix. The authors conducted extensive experiments on various models and datasets, demonstrating that TopLoRA consistently outperforms LoRA and its variants. At the same rank, TopLoRA achieves a 2-3% accuracy improvement over LoRA.

**Questions:**

Are there any evidence that support the authors' claim "Existing methods fail to capture token-specific information"?

**Ethical Concerns:**

["NO or VERY MINOR ethics concerns only"]

**Final Justification:**

The author response provides extensive experiments which generally address my concerns. Therefore I updated my score to 4.

**Limitations:**

yes

**Quality:**

3

**Strengths And Weaknesses:**

**Strengths**

(1) This paper is well motivated by considering token-wise differences, and the proposed method TopLoRA is clear and makes sense to me.

(2) The proposed method is comprehensively evaluated on multiple tasks (General Language Understanding, Mathematical Reasoning, Commonsense Reasoning) on multiple LLMs, and has performance improvement compared to baseline methods.

(3) Some theoretical analysis is shown to make the motivation more convincing.

**Weaknesses**

(1) The authors claim that their method achieves 2-3% accuracy improvement over LoRA. Firstly, if this comparison is between TopLoRA and LoRA with the same rank, then it will be unfair to LoRA as TopLoRA leads to ~50% trainable parameter increase. Under a fair setting, the improvement of TopLoRA will shrink. In addition, if we compare TopLoRA to stronger LoRA variants, the improvement is smaller or even marginal.

(2) In L113, the authors claim that “Existing methods that apply a uniform input-output projection across all tokens fail to capture token-specific information.” But in my opinion, the original Q,K,V mapping is capturing token-wise information, so the attached updated LoRA layers are capable of capturing token-wise information as well. It seems to me that TopLoRA is actually providing an additional channel that strengthens the token-wise information. In such cases, the methodological contribution of TopLoRA is limited to me.

---

> ### Author Rebuttal · Authors · 2025-07-30
>
> **Hi Reviewer 6dXU:**
>
> Thank you for your recognition of our **motivation**, **methodological clarity**, and **extensive experiments**. We notice that the concerns mainly relate to the **additional parameter cost of TopLoRA** and its **conceptual distinction from attention**. We have carefully addressed both points through further clarification and empirical evidence. Below, we provide responses to each point individually.
>
> ***`Q1: "Unfair comparison between LoRA and TopLoRA."`***
>
> **R1:**
> Due to the improved structure, TopLoRA introduces approximately 50% more parameters than LoRA at the same rank.
> However, the performance gains it achieves significantly surpass those obtained by merely increasing the rank of LoRA.
> As noted in line 227, "these gains surpass those achieved by increasing LoRA’s rank by a factor of four".
> For example, on Mathematical Reasoning tasks with Gemma-7B, LoRA (r=32) requires **19.3M** trainable parameters to achieve an accuracy of **72.71**, whereas TopLoRA (r=8) achieves a higher accuracy of **73.11** with only **6.88M** parameters.
>
> To ensure a fair comparison, we evaluated TopLoRA (rank=5/10/20) and compared it with LoRA (rank=8/16/32). In these settings, they use a similar number of parameters. As shown below, we report the average accuracy on mathematical and commonsense reasoning tasks. The *Accuracy Gain* column indicates the performance gain of TopLoRA over LoRA under comparable parameter budgets.
> **TopLoRA achieves a 1–3% boost in accuracy with a similar number of parameters**. Notably, increasing the LoRA rank by a factor of four yields only a modest 1–2% improvement.
>
> |||#Params|Mathematical|Accuracy Gain|Commonsense|Accuracy Gain|
> |-|-|-|-|-|-|-|
> |Gemma-7B|LoRA($r=8$)|4.82M|71.44|-|85.95|-|
> ||TopLoRA($r=5$)|4.30M|72.97|**1.53**|87.68|**1.73**|
> ||LoRA($r=16$)|9.63M|72.17|-|86.43|-|
> ||TopLoRA($r=10$)|8.60M|73.16|**0.99**|87.92|**1.49**|
> ||LoRA($r=32$)|19.3M|72.71|-|86.61|-|
> ||TopLoRA($r=20$)|17.2M|74.14|**1.43**|88.44|**1.83**|
> |Llama-3-8B|LoRA($r=8$)|4.72M|68.19|-|85.45|-|
> ||TopLoRA($r=5$)|4.92M|70.94|**2.75**|86.37|**0.92**|
> ||LoRA($r=16$)|9.44M|70.19|-|86.15|-|
> ||TopLoRA($r=10$)|9.83M|72.37|**2.18**|86.68|**0.53**|
> ||LoRA($r=32$)|18.9M|71.67|-|86.72|-|
> ||TopLoRA($r=20$)|19.7M|73.52|**1.85**|87.92|**1.20**|
> |Qwen2.5-14B|LoRA($r=8$)|8.65M|79.09|-|91.24|-|
> ||TopLoRA($r=5$)|9.09M|80.11|**1.02**|91.60|**0.36**|
> ||LoRA($r=16$)|17.3M|79.43|-|91.51|-|
> ||TopLoRA($r=10$)|18.2M|80.87|**1.44**|91.93|**0.42**|
> ||LoRA($r=32$)|34.6M|80.15|-|91.69|-|
> ||TopLoRA($r=20$)|36.4M|81.25|**1.10**|92.85|**1.16**|
>
> On the other hand, inspired by Reviewer SWr9, we explored **sharing a common set of learnable projectors across different layers** for weight modulation. The results show that **a small number of shared projectors—approximately 8—is sufficient to achieve performance comparable to the original TopLoRA**. This strategy offers an effective means of further reducing the number of trainable parameters in TopLoRA without sacrificing accuracy. Please kindly check **Q5** in our response to Reviewer **SWr9** for experimental results.
>
> ***`Q2: "If we compare TopLoRA to stronger LoRA variants, the improvement is smaller or even marginal."`***
>
> **R2:**
> Compared to stronger LoRA variants, our approach still achieves substantial improvements.
>
> Our paper benchmarks several state-of-the-art methods, including MELoRA, DoRA, and HydraLoRA. The results show that TopLoRA achieves higher accuracy while using fewer parameters. Please kindly check Tables 1/2/3 in the paper and compare TopLoRA (r=8) with DoRA (r= 16), MoELoRA (r=16), and HydraLoRA (r=8), respectively.
>
> We also include comparisons with other strong baselines here, such as the higher rank methods HiRA and KronA, and the dynamic rank adjustment method AdaLoRA. Note that KronA does not have a rank hyperparameter, and its number of trainable parameters is fixed and non-adjustable.
> Due to their architectural modifications, HiRA and KronA require larger learning rates to perform well. Accordingly, we tuned the learning rate over {1e-4, 3e-4, 1e-3, 3e-3, 1e-2} and report the best results. AdaLoRA is configured with an initial rank of 24 and a target rank of 16. TopLoRA uses a rank of 10 to maintain similar parameter counts as the baselines.
>
> As shown below, the *Accuracy Gain* column reflects performance gains relative to standard LoRA.
> KronA shows inferior accuracy compared to LoRA, likely due to its extremely small number of trainable parameters. Nevertheless, TopLoRA consistently outperforms all of these state-of-the-art methods under comparable parameter budgets.
>
> Importantly, as shown in the table in **R1**, doubling or even quadrupling the LoRA rank does not lead to accuracy gains comparable to those achieved by TopLoRA—gains that these baselines fail to match. The improvements delivered by TopLoRA are substantial, underscoring the effectiveness of its architectural design.
>
>
> |||#Params|Mathematical|Accuracy Gain|Commonsense|Accuracy Gain|
> |-|-|-|-|-|-|-|
> |Gemma-7B|LoRA($r=16$)|9.63M|72.17|-|86.43|-|
> ||KronA|0.60M|69.46|**-2.71**|84.49|**-1.94**|
> ||HiRA($r=16$)|9.63M|72.91|**0.74**|87.09|**0.66**|
> ||AdaLoRA($r=16$)|9.63M|72.65|**0.48**|86.98|**0.55**|
> ||TopLoRA($r=10$)|8.60M|73.16|**0.99**|87.92|**1.49**|
> |Llama-3-8B|LoRA($r=16$)|9.44M|70.19|-|86.15|-|
> ||KronA|0.52M|67.41|**-2.78**|83.97|**-2.18**|
> ||HiRA($r=16$)|9.44M|70.96|**0.77**|86.38|**0.23**|
> ||AdaLoRA($r=16$)|9.44M|70.50|**0.31**|86.20|**0.05**|
> ||TopLoRA($r=10$)|9.83M|72.37|**2.18**|86.68|**0.53**|
> |Qwen2.5-14B|LoRA($r=16$)|17.3M|79.43|-|91.51|-|
> ||KronA|0.95M|78.73|**-0.70**|89.36|**-2.15**|
> ||HiRA($r=16$)|17.3M|79.59|**0.16**|91.78|**0.27**|
> ||AdaLoRA($r=16$)|17.3M|79.65|**0.22**|91.49|**-0.02**|
> ||TopLoRA($r=10$)|18.2M|80.87|**1.44**|91.93|**0.42**|
>
> ***`Q3: "The original Q,K,V mapping is capturing token-wise information, so the attached updated LoRA layers are capable of capturing token-wise information as well ... In such cases, the methodological contribution of TopLoRA is limited to me."`***
>
> **R3:**
> This is an interesting question​​, and we appreciate the opportunity to clarify the distinction between token interactions (handled by attention) and token-specific processing (our focus within LoRA layers), as well as why TopLoRA represents a methodological innovation.
>
> ​**1. Roles of Attention vs. LoRA:**
> The QKV mappings in attention are designed to capture inter-token dependencies (how tokens influence one another). In contrast, LoRA layers operate prior to attention, transforming individual token representations. The ability of LoRA to preserve token-specific information is therefore orthogonal to the function of attention.
>
> **2. ​​Limitation of Standard LoRA​​:**
> As noted in Eq.(1) in our paper, LoRA weights $BA=Q_BPQ_A$​ projects inputs into a fixed subspace (spanned by $Q_A$). This means that token information orthogonal to $Q_A$ is irrecoverably lost in the transformation. Increasing rank (more $Q_A$ columns) expands this subspace but remains static across tokens, and it cannot adapt to token-specific features.
>
> **3. TopLoRA’s Innovation​​:**
> Instead of expanding rank, TopLoRA generates token-dependent diagonal matrices $\Sigma_X$ to dynamically modulate the input-output mapping per token. This token-aware reweighting enriches the expressiveness of LoRA by allowing the transformation to adapt to each token’s unique features.
>
> **​4. Why This Matters for Attention​​:**
> Better token-specific transformations (via TopLoRA) lead to higher-quality QKV representations. Attention can only model inter-token relationships effectively if the underlying QKV mappings already capture rich, discriminative token features.
>
> ​​In summary​​, TopLoRA is not just "another channel" for token-wise information but a fundamental enhancement to LoRA’s capacity to preserve token-specific features—a prerequisite for attention to function optimally. This is a distinct contribution beyond static low-rank adaptations.
>
>
> ***`Q4: "Evidence that support the claim 'Existing methods fail to capture token-specific information'"`***
>
> **R4:**
> As noted in Eq.(1) of our paper, the standard LoRA formulation $BA=Q_BPQ_A$projects token representations into a fixed low-dimensional subspace spanned by $Q_A$. This inherently limits the model’s ability to capture token-specific variations: any input component orthogonal to this subspace is irrecoverably lost. Consequently, when the rank is small—as is often necessary for efficiency—the amount of discarded token-specific information increases.
>
> To further quantify this limitation, we compute the **mutual information (MI)** between input tokens $X$ and output representations $Y$ using the InfoNCE lower bound [1], a standard technique in representation learning. Each input-output pair $(X_i,Y_i)$ is treated as positive, with others as negatives. A higher MI indicates better preservation of token-specific information. This method allows us to provide a tractable, quantitative comparison of how much token-level information is retained.
>
> Due to the image upload limitations during rebuttal, we summarize a subset of the results on LLaMA-3-8B (Math) in the table below and will include full visualizations and analysis in the revised paper.
>
> ||Layer5|Layer10|Layer20|
> |-|-|-|-|
> |LoRA($r=8$)|1.13|0.62|1.96|
> |LoRA($r=16$)|1.56|0.76|2.12|
> |TopLoRA($r=8$)|2.31|1.02|2.97|
>
> These results show that **TopLoRA retains significantly more mutual information than standard LoRA**, even at smaller ranks, confirming its stronger ability to capture token-specific features. Together with the consistent performance improvements shown in our experiments, this provides strong support for our central claim: TopLoRA’s dynamic modulation enhances the expressiveness of LoRA.
>
> [1] A. van den Oord et al., Representation Learning with Contrastive Predictive Coding, 2018.

---

> > ### Comment · Reviewer_6dXU · 2025-08-05
> >
> > Thank you for your response, which addresses most of my concerns. I have therefore increased my score.

---

> > > ### Author Response · Authors · 2025-08-06
> > > **Sincere Thanks for Your Feedback and Support**
> > >
> > > Thank you for your thoughtful review and for raising the score. We sincerely appreciate your feedback. We will incorporate the additional discussions and explanations, along with the new experimental results, in the revised version. Many thanks again for your support.

---

### Official Review · Reviewer_2KRw · 2025-07-02

**Clarity:** 4
**Significance:** 3
**Originality:** 4
**Rating:** 5
**Confidence:** 5

**Summary:**

This paper aims to improve the performance of Low-Rank Adaptation (LoRA) for fine-tuning large language models. The key insight is that the limitations of standard LoRA stem not only from its rank (which determines input/output space dimensionality) but also from its shared input-output projection across all tokens. TopLoRA addresses this by dynamically adjusting the LoRA weights ($B\Sigma_X A$) for each input token $X$ through learned diagonal matrices $\Sigma_X$. It enables token-wise projections while maintaining the original rank. TopLoRA is evaluated on a range of tasks and models, clearly demonstrating its advantages.

**Questions:**

The paper mentions applying Exp and RMSNorm to diagonal matrices, but their role needs further explanation. Also, does the order in which they are applied matter?

How does TopLoRA perform on vision tasks?

**Ethical Concerns:**

["NO or VERY MINOR ethics concerns only"]

**Final Justification:**

Thank the authors for their detailed response. My concerns have been well addressed through the additional clarifications and new experimental results.

Several reviewers, including myself, suggested adding more baselines (e.g., LoRA+, KronA, HiRA, AdaLoRA, and MoELoRA). I appreciate the authors’ significant efforts in this regard. The method’s orthogonality to existing approaches is now clearly demonstrated, and the inclusion of these baselines provides stronger empirical validation, showing that TopLoRA consistently outperforms these competitive methods. The experimental results are now more comprehensive and convincing.

Accordingly, I raise my score to 5. This paper makes a valuable contribution to the field, and I recommend acceptance.

**Paper Formatting Concerns:**

NA.

**Quality:**

3

**Strengths And Weaknesses:**

Strengths:
- This paper is well written and easy to understand. It provides a clear framework to understand LoRA through QR/LQ decomposition, showing how to view LoRA weights from the perspective of input space, output space, and input-output projection.
- TopLoRA does not follow the common method of increasing rank, but focuses on enhancing projection diversity. Its token-wise diagonal matrix $\Sigma_X$ is simple and powerful, and can improve expressiveness without increasing rank. This is very inspiring for the optimization of LoRA.
- The experiments cover three model sizes (7 B, 8 B, 14 B), multiple architectures (RoBERTa, Gemma, LLaMA-3, Qwen 2.5), multiple tasks (GLUE, mathematical reasoning, common sense reasoning), multiple ranks and strong baselines (DoRA, MELoRA, HydraLoRA). Compared with LoRA, the accuracy continues to increase by 2-4% at the same rank, clearly proving the effectiveness of TopLoRA.

Weakness:

Although MoELoRA is mentioned (in Section 3.3), only its variant HydraLoRA is compared. A detailed comparison with MoELoRA is needed as the two are very similar.

---

> ### Author Rebuttal · Authors · 2025-07-30
>
> **Hi, Reviewer 2KRw:**
>
> Thank you for your positive assessment of the **clarity** and **novelty** of our paper. In response to the concerns regarding comparisons on **vision tasks** and evaluations against **MoELoRA**, we have conducted substantial supplementary experiments and analyses to strengthen our findings. Please find our point-by-point responses below.
>
> ***`Q1: "Comparison with MoELoRA (and other methods)."`***
>
> **R1:**
> As suggested, we conducted a more in-depth investigation of MoELoRA. Specifically, we added experiments using MoELoRA with four fully activated experts, each implemented as a rank-4 LoRA module. This configuration yields a total parameter count comparable to that of a rank-16 TopLoRA.
>
> As shown below, we report the average accuracy of different models on mathematical and commonsense reasoning benchmarks. The *Accuracy Gain* column reflects the performance improvement over LoRA ($r=16$).
> The results show that TopLoRA significantly outperforms MoELoRA under a similar parameter budget.
>
> Furthermore, we replaced each expert in MoELoRA with a rank-4 TopLoRA module, creating a variant we refer to as MoETopLoRA. Our findings indicate that the performance gains from MoELoRA and TopLoRA are largely additive, resulting in further improvements.
>
> In addition, we have included more relevant baseline comparisons, such as the rank-increasing methods HiRA and KronA, the dynamic rank adjustment method AdaLoRA, and the optimization-enhanced method LoRA+. For further details, please refer to **Q1** in our response to Reviewer **SWr9** and **Q5** in our response to Reviewer **2FJn**. Thank you again for your thoughtful feedback.
>
> |  |  | #Params | Mathematical | Accuracy Gain | Commonsense | Accuracy Gain  |
> |---|---|---|---|---|---|---|
> | Gemma-7B | LoRA ($r=16$) | 9.63M | 72.17 | - | 86.43 | -  |
> |  | MoELoRA ($r=4,n=4$) | 10.7M | 72.3 | **0.13** | 86.80 | **0.37**  |
> |  | TopLoRA ($r=16$) | 13.8M | 73.28 | **1.11** | 87.45 | **1.02**  |
> |  | MoETopLoRA ($r=4,n=4$) | 14.9M | 73.49 | **1.32** | 87.62 | **1.19**  |
> |  | LoRA ($r=32$) | 19.3M | 72.71 | **0.54** | 86.61 | **0.18**  |
> | Llama-3-8B | LoRA ($r=16$) | 9.44M | 70.19 | - | 86.15 | -  |
> |  | MoELoRA ($r=4,n=4$) | 11.0M | 71.36 | **1.17** | 86.42 | **0.27**  |
> |  | TopLoRA ($r=16$) | 15.7M | 72.95 | **2.76** | 86.84 | **0.69**  |
> |  | MoETopLoRA ($r=4,n=4$) | 17.2M | 73.33 | **3.14** | 87.28 | **1.13**  |
> |  | LoRA ($r=32$) | 18.9M | 71.67 | **1.48** | 86.72 | **0.57**  |
> | Qwen2.5-14B | LoRA ($r=16$) | 17.3M | 79.43 | - | 91.51 | -  |
> |  | MoELoRA ($r=4,n=4$) | 20.3M | 79.83 | **0.40** | 91.44 | **-0.07**  |
> |  | TopLoRA ($r=16$) | 29.1M | 80.76 | **1.33** | 91.71 | **0.20**  |
> |  | MoETopLoRA ($r=4,n=4$) | 32.1M | 80.65 | **1.22** | 91.73 | **0.22**  |
> |  | LoRA ($r=32$) | 34.6M | 80.15 | **0.72** | 91.69 | **0.18**  |
>
> ***`Q2: "The role (of Exp and RMSNorm) needs further explanation. Also, does the order in which they are applied matter?"`***
>
> **R2:**
> To clarify this, we first highlight three key points:
>
> 1. TopLoRA extends LoRA by introducing a diagonal matrix $\Sigma_X$ in the form of $B\Sigma_XA$.
> $\Sigma_X$ is generated by a learnable projector based on the input token $X$. Notably, LoRA is a special case where $\Sigma_X$ is the identity matrix.
> 1. When the standard deviation of the parameters in $\Sigma_X$ is small (i.e., the values are tightly concentrated), $\Sigma_X$ becomes nearly identical across different tokens, limiting its ability to modulate token-specific weights.
> 2. If an element in $\Sigma_X$ is zero or close to zero, it is equivalent to setting the column of B or the row of A to zero. If there are a large number of values distributed near zero, it may reduce the expressive power of TopLoRA.
>
> Given that the standard deviation of the projector output is typically small (around 0.05), we first apply RMSNorm to scale $\Sigma_X$, removing the effects of both input token variance and projector weight variance.
>
> Additionally, since the projector output tends to have a mean close to zero, we use an exponential (Exp) transformation to shift the distribution of $\Sigma_X$ from around 0 to around 1.
>
> Note that RMSNorm does not include a mean subtraction step (or implicitly assumes zero mean). Therefore, if we apply the Exp transformation before normalization, we must use LayerNorm instead of RMSNorm to account for the shift in mean. Continuing to use RMSNorm after Exp would fail to properly normalize the standard deviation of $\Sigma_X$.
>
> We also added experimental results to compare the effects of different orders of applying the exponential function (Exp) and normalization methods (RMSNorm and LayerNorm). Based on Table 4 in the paper, we introduced two additional settings where Exp is applied first, followed by either RMSNorm or LayerNorm. The results show that there is no significant difference between RMSNorm(Exp($\cdot$)) and simply applying Exp($\cdot$) alone. In contrast, LayerNorm(Exp($\cdot$)) achieves performance close to the default setting, Exp(RMSNorm($\cdot$)).
>
> | TopLoRA ($r=8$) / Mathematical  | Gemma-7B | Llama-3-8B | Qwen2.5-14B  |
> |---|---|---|---|
> | Exp(RMSNorm($\cdot$))  (Default Setting) | 73.11 | 71.84 | 80.52  |
> | RMSNorm($\cdot$) | 72.84 | 70.56 | 79.98  |
> | Exp($\cdot$) | 72.69 | 71.77 | 80.23  |
> | RMSNorm(Exp($\cdot$)) | 72.73 | 71.36 | 80.07  |
> | LayerNorm(Exp($\cdot$)) | 72.97 | 71.63 | 80.58  |
>
> ***`Q3: "How does TopLoRA perform on vision tasks?"`***
>
> **R3:**
> As suggested, we have included fine-tuning results of TopLoRA on an image classification task. Specifically, we fine-tuned a pretrained ViT model (google/vit-base-patch16-224-in21k) on the CIFAR-100 dataset for 60 epochs. The learning rate was tuned over {1e-4, 3e-4, 1e-3, 3e-3}, with the final value set to 3e-4. The rank was varied among 8, 16, 32, and 64. Both LoRA and TopLoRA were applied to the query, value, and classifier weights.
> As shown below, TopLoRA achieves comparable accuracy while reducing the number of trainable parameters by approximately 6×.
>
> |  | LoRA #Params | LoRA Accuracy | TopLoRA #Params | TopLoRA Accuracy | Accuracy Gain  |
> |---|---|---|---|---|---|
> | rank=8 | 0.30M | 88.95 | 0.45M | 90.20 | **1.25**   |
> | rank=16 | 0.60M | 89.96 | 0.91M | 91.16 | **1.20**   |
> | rank=32 | 1.20M | 90.12 | 1.82M | 91.17 | **1.05**   |
> | rank=64 | 2.41M | 90.26 | 3.64M | 91.60 | **1.34**   |

---

> > ### Comment · Reviewer_2KRw · 2025-08-03
> >
> > Thank the authors for their detailed response. My concerns have been well addressed through the additional clarifications and new experimental results.
> >
> > Several reviewers, including myself, suggested adding more baselines (e.g., LoRA+, KronA, HiRA, AdaLoRA, and MoELoRA). I appreciate the authors’ significant efforts in this regard. The method’s orthogonality to existing approaches is now clearly demonstrated, and the inclusion of these baselines provides stronger empirical validation, showing that TopLoRA consistently outperforms these competitive methods. The experimental results are now more comprehensive and convincing.
> >
> > Accordingly, I raise my score by 1 point. This paper makes a valuable contribution to the field, and I recommend acceptance.

---

> > > ### Author Response · Authors · 2025-08-04
> > > **Sincere Thanks for Your Feedback and Support**
> > >
> > > Thank you again for your thorough review and constructive feedback. We’re pleased that our clarifications and experiments resolved your concerns. We greatly appreciate your recognition and the raised score—your comments significantly helped improve the paper.

---

### Decision · Program_Chairs · 2025-09-17

**Decision:**

Accept (poster)

**Comment:**

This paper received all positive reviews. After discussion, all reviewers agreed to accept this paper.  The authors are encouraged to put the following new results and discussions into the camera-ready version.
1. Add more baselines (e.g., LoRA+, KronA, HiRA, AdaLoRA, and MoELoRA) in the experiments.
2. Add a more formal and thorough discussion of related works.